

# Retrieval and Validation of Total Seasonal Liquid Water Amounts in the Percolation Zone of Greenland Ice Sheet Using L-band Radiometry

Alamgir Hossan[1], Andreas Colliander[1], Baptiste Vandecrux[2], Nicole-Jeanne Schlegel[3], Joel Harper[4], Shawn Marshall[5], Julie Z Miller[6]

[1]Jet Propulsion Laboratory, California Institute of Technology
[2]Geological Survey of Denmark and Greenland
[3]NOAA/OAR Geophysical Fluid Dynamics Laboratory (GFDL)
[4]Department of Geosciences, University of Montana
[5]Department of Geography, University of Calgary
[6]Cooperative Institute for Research in Environmental Sciences, University of Colorado Boulder

*Correspondence to*: Alamgir Hossan (alamgir.hossan@jpl.nasa.gov) and Andreas Colliander (andreas.colliander@jpl.nasa.gov)

**Abstract.** Quantifying the total liquid water amounts (LWA) in the Greenland ice sheet (GrIS) is critical for understanding GrIS firn processes, mass balance, and global sea-level rise. Although satellite microwave observations are very sensitive to ice sheet melt and thus can provide a way of monitoring the ice sheet melt globally, estimating total LWA, especially the sub-surface LWA, remains a challenge. Here, we present a microwave retrieval of LWA over Greenland using enhanced resolution L-band brightness temperature (TB) data products from the NASA Soil Moisture Active Passive (SMAP) satellite for the 2015-2023 period. L-band signals receive emission contributions deep in the ice sheet and are sensitive to the liquid water content (LWC) in the firn column. Therefore, they can estimate the surface-to-subsurface LWA, unlike higher frequency signals (e.g., 18 and 37 GHz bands), which are limited to the top few centimeters of the surface snow during the melt. We used vertically polarized TB (V-pol TB) with empirically derived thresholds to detect liquid water and identify distinct ice sheet zones. A forward model based on radiative transfer in the ice sheet was used to simulate TB. The simulated TB was then used in an inversion algorithm to estimate LWA. Finally, the retrievals were compared with the LWA obtained from two sources. The first source was a locally calibrated ice sheet energy and mass balance (EMB) model, which was forced by *in situ* measurements from six automatic weather stations (AWS) of the Programme for Monitoring of the Greenland Ice Sheet (PROMICE) and Greenland Climate Network (GC-Net) located in the percolation zone of the GrIS. The second source was the corresponding LWA obtained from the Glacier Energy and Mass Balance (GEMB) model within the National Aeronautics and Space Administration's (NASA) Ice-sheet and Sea-Level System Model (ISSM). The retrievals show generally good





agreement with both the references, demonstrating the potential for advancing our understanding of ice sheet physical
processes to better project Greenland's contribution to the global sea level rise in response to the warming climate.
**1 Introduction**
Continuous mass loss of the Greenland Ice Sheet (GrIS) has been a significant concern in the context of climate change and
associated sea level rise (Khan et al., 2015; Mouginot et al., 2019; Otosaka et al., 2023; Shepherd et al., 2020). Greenland has
lost about 330 billion tons of mass, equivalent to around 1 mm global sea level rise, per year on average for the last two decades
(Greene et al., 2024; Khan et al., 2022). This will likely accelerate in the coming decades, even with the most optimistic
warming scenario.
Mass loss occurs through surface melt and the subsequent runoff of meltwater towards the ice sheet margin and solid
ice discharge (calving) at marine-terminating outlet glacier termini. While meltwater runoff has been the dominant contributor
to mass loss in Greenland, both have increased in the last few decades (Van Den Broeke et al., 2016; Greene et al., 2024;
Mouginot et al., 2019; Vandecrux et al., 2023a). In the ablation area, the winter snowpack is melted out every summer and the
meltwater enter an efficient drainage network of streams and lakes toward the margin (Smith et al., 2017). Higher up on the
ice sheet, in the accumulation area, there is less melt, and a porous snow layer accumulated over the years, called firn, leads to
the percolation and refreezing of surface melt, buffering additional sea level rise (Harper et al., 2012; Samimi et al., 2020).
However, with intense and frequent melt events, thick ice layers, called ice slabs, are formed from meltwater refreezing,
impeding vertical percolation of meltwater and promoting horizontal runoff (Culberg et al., 2021; Jullien et al., 2023;
MacFerrin et al., 2019; Miller et al., 2022b, 2020b; Tedstone and Machguth, 2022). Increased refreezing resulted in a loss of
approximately 5% of GrIS firn air content (FAC) between 1996 and 2019 (Medley et al., 2022). These effects gradually
diminish the ice sheet's inherent capability to retain meltwater and buffer sea level rise (Harper et al., 2012; Mikkelsen et al.,
2016; Vandecrux et al., 2019).
Furthermore, increased melting contributes to forming supraglacial, englacial, and subglacial meltwater features (e.g.,
lakes, rivers, slush, crevasses, moulins, and firn aquifers, etc.) that can augment dynamical discharge and calving losses by
lubricating the basal sliding surface and accelerating the flow of outlet glaciers (Hoffman et al., 2011; Schoof, 2010; Sundal
et al., 2011; Zwally et al., 2002). Therefore, meltwater not only contributes to sea-level rising through direct runoff, but it can
also alter the physical structure that governs the dynamics and evolution of the ice sheet. Hence, quantification of total surface
and subsurface liquid water is essential to understand ice sheet response to climate changes and project sea level rise accurately.
Surface melt and liquid water amount (LWA) can be estimated with various techniques. *In situ* AWS networks
provide meteorological observations (Fausto et al., 2021), which can drive surface energy and mass balance (EMB) models to
derive surface melt and LWA. Other *in situ* measurements such as upward-looking radar (Heilig et al., 2018) or time domain
resistivity probes(Samimi et al., 2021) can also be used to measure LWA at a given site. Due to logistical constraints, these
point observations have a limited spatial and temporal coverage.





Regional climate models (RCM) are primarily used to estimate ice sheet-wide LWA, surface mass balance (SMB),
and their changes (Fettweis et al., 2020). The results of RCMs are difficult to validate on the scale of the ice sheet, given the
scarcity of *in situ* data to constrain and calibrate these models. Moreover, diversity exists in representation of the surface and
sub-surface firn processes among RCMs, leading to significant uncertainties in LWA estimates and their temporal and spatial
variabilities (Fettweis et al., 2020; Thompson-Munson et al., 2023; Vandecrux et al., 2020; Verjans et al., 2019).
Satellite-based observations, especially microwave sensors, are very sensitive to ice sheet melting, manifested by
large changes in dielectric constant with liquid water and can provide global coverage in day-night and all-weather conditions
(Abdalati and Steffen, 1997; Mote and Anderson, 1995; Picard et al., 2022; Tedesco, 2007; Tedesco et al., 2007; Zwally and
Fiegles, 1994). Accordingly, both active (radars) and passive sensors (radiometers) have been used to monitor surface melting
across Greenland and Antarctica ice sheets (Abdalati and Steffen, 1995, 2001; Hall et al., 2009; Mote, 2007; Nghiem et al.,
2001; Tedesco, 2007; Wismann, 2000; Zwally and Fiegles, 1994). However, these conventional approaches applying high-
frequency bands (i.e., 18 and 36 GHz) from the legacy and operational radiometers (Abdalati and Steffen, 1997; Ashcraft and
Long, 2006; Colosio et al., 2021; Fettweis et al., 2007, 2011; Tedesco, 2007, 2009; Tedesco et al., 2007; Zwally and Fiegles,
1994) can only track the surface and near-surface binary melt status, not the meltwater propagation into the deeper layers
because of their limited penetration depth and sensitivity to LWC (Colliander et al., 2022a, b, 2023; Mousavi et al., 2022). The
emergence of L-band (1 - 2 GHz) radiometry, marked by the launch of ESA's Soil Moisture and Ocean Salinity (SMOS)
mission (November 2009 - present) and the collaborative effort between NASA and Argentina's space agency CONAE in the
Aquarius mission (October 2011 - June 2015), followed by NASA's Soil Moisture Active Passive (SMAP) mission (March
2015 - present), has opened up the possibilities for monitoring ice sheet meltwater at greater depths. L-band signals can
penetrate deeper and provide a more accurate estimate of sub-surface liquid water (Colliander et al., 2022b; Miller et al., 2020a,
2022a, b; Mousavi et al., 2022). Nevertheless, only a few attempts have been made to quantify the amount of liquid water
(Colliander et al., 2022a; Houtz et al., 2019, 2021; Mousavi et al., 2021; Schwank and Naderpour, 2018). In this paper, we use
multi-year L-band observations from SMAP to quantify the LWA on the GrIS and examine their spatial and temporal
variability.
**2 Methods**
**2.1 SMAP L-band Enhanced Resolution Brightness Temperatures**
SMAP was launched on January 31, 2015, and has been operational since March 31, 2015 (Entekhabi et al., 2010). It was
placed in a 685-km altitude and 98.1∘ inclination sun-synchronous polar orbit with 6 AM/6 PM equator-crossing. It carries a
conically scanning radiometer operating in 1.41 GHz (L-band) with a constant incidence angle of 40∘ that results 1000-km
wide swath giving twice daily coverage of GrIS. It measures brightness temperature (TB) in fully polarimetric mode giving
the horizontal and vertical polarizations, as well as the 3rd and 4th Stokes parameters with native 38-km spatial resolution.





The radiometric precision of the SMAP radiometer is within 0.5 K (Chaubell et al., 2018, 2020; Piepmeier et al., 2017). For
Jun 20 – July 23, 2019, and Aug 6 - Oct 16, 2022, SMAP does not have results because of an operational outage of the satellite.
Here, we used SMAP L-band enhanced-resolution TB products generated using the radiometer form of the
Scatterometer Image Reconstruction (rSIR) algorithm and projected on the EASE-2 3.125 km grid (Brodzik et al., 2021; Long
et al., 2019). The rSIR algorithm leverages the measurement response function (MRF) of each observation and combines the
overlapping MRFs to reconstruct enhanced-resolution TB images. The effective resolution of SMAP enhanced-resolution TB
products posted on a 3.125 km grid is ~30 km compared to the 36 km effective resolution of the SMAP original data products
(Long et al., 2023). The data product provides two TB images daily – the morning and evening facilitating the resolution of
diurnal variability. The spatial oversampling and resolution enhancement enables an improved characterization of spatial
heterogeneity (Long et al., 2023). The land–ocean mask used to locate the ice sheet edge comes from PROMICE (Citterio and
Ahlstrøm, 2013).
**2.2 Microwave Radiometric Response of GrIS**
**2.2.1 Theoretical Background**
Microwave radiometers measure the naturally emitted thermal radiation, called the brightness temperature (TB), by the firn as
observed in the microwave portion of the electromagnetic spectrum. It is related to the emissivity $e$ and the effective physical
temperature $T_{phy}$ of snow/firn/ice media for a given frequency $f$, polarization $p$, and incidence angle $\theta$. If firn were vertically
homogeneous or isothermal, the TB could be found according to Rayleigh-Jeans approximation (Ulaby and Long, 2014):
$T_B(f, p, \theta) = e\, T_{phy}$ (1)
However, firn is not a vertically homogenous medium. Both the emissivity and temperature vary with depth. As a
result, the TB is given by a depth-integrated product of physical temperature and emissivity, weighted by the emissive,
absorptive, and scattering properties of the snow/firn/ice layers (Jay Zwally, 1977) which is strongly dependent on the
frequency of observation.
To account for the depth dependencies of snow and ice properties, firn is considered as a complex multilayer dense
medium. For each layer, an effective physical temperature and permittivity is determined from firn absorptive and scattering
properties. Then the microwave emission and its propagation are typically modeled using equation of radiative transfer (RT).
Considering firn as a stack of N layers consisting of isotropic and homogeneous material in each layer, the RT equation can
be given as (Jin, 1994, 1997; Picard et al., 2013; Tsang et al., 2000):
$\cos\theta \frac{d}{dz} \boldsymbol{T_B}(z, \theta, \emptyset) = \kappa_a \boldsymbol{T}(z)\boldsymbol{I} - \kappa_e \boldsymbol{T_B}(z, \theta, \emptyset) + \int_0^{\frac{\pi}{2}} \int_0^{2\pi} \sin\theta'\, d\theta' d\emptyset' \boldsymbol{P}(\theta, \emptyset, \theta', \emptyset') \boldsymbol{T_B}(z, \theta', \emptyset')$ (2)
Here, $\boldsymbol{T_B}(z, \theta, \emptyset)$ denotes the vertically and horizontally polarized brightness temperatures at depth $z$ propagating along a
direction characterized by $\theta$ (zenith angle) and $\phi$ (azimuth angle). $\kappa_e$, $\kappa_a$, and $\kappa_s$ are the extinction, absorption, and scattering



coefficients, respectively, representing medium properties. For an isotropic medium, the extinction coefficient can be described
as, $\kappa_e = \kappa_a + \kappa_s$. $\theta'$ and $\phi'$ are slant angles and $\boldsymbol{P}$ is bistatic scattering phase function. $\boldsymbol{T}(z)$ is the physical temperature of
snow at depth z, and $\boldsymbol{I}$ is a unit vector. Thus, the first term on the right-hand side of Eq. 2 represents the microwave emission
$(T_B)$ of snow/fir/ice from depth z, and the second term denotes the extinction (attenuation) of the emission due to absorption
and scattering. The third term represents the sum of total scattered emission in the direction of the receiver (as specified by $\theta$
and $\phi$). Eq. 2 is solved analytically or numerically subject to boundary conditions at each layer interface and at the top and
bottom of the medium

The extinction coefficient, $\kappa_e$ is function of the effective dielectric constant of the layer and frequency of the

observation. Thus, the overall TB is given by the depth-integrated profiles of the effective physical temperature and dielectric
constant of each layer. So, penetration depth plays a key role in determining the variability of TB, especially in low-frequency
bands. For a low-loss media like firn, the penetration depth can be approximated as (Elachi and Zyl, 2021):
$\delta = \frac{c\sqrt{\epsilon'}}{2\pi f \epsilon''}$                                                                                           (3)
where c is the speed of light and, $\epsilon'$ and $\epsilon''$ are the real and imaginary parts of the dielectric constant of the firn. As shown, $\delta$
is inversely proportional to both, $f$ and $\epsilon''$. L-band signal thus penetrates a significantly thicker layer than the higher frequency,
like Ka-band signal. Liquid water markedly increases $\epsilon''$ (compared to $\sqrt{\epsilon'}$), decreasing the penetration depth for any frequency.

There are two types of scattering processes in the snow/firn medium affecting the propagation: surface scattering and

volume scattering. The relative size of the scatterers compared to the wavelength determines the degree and types of scattering.
For high frequency bands (>10GHz), the impact of volume scattering is critical because the fractional volume of scatterers
(snow/firn) is significant. This is why the high-frequency signals interact more with fresh snow, grain size, and roughness at
the surface. Low-frequency signals (<10GHz) are relatively insensitive to volume scattering from snow grains because the size
of the scatterers is much smaller than the wavelength. Surface scattering occurs due to surface irregularities at the interface
between layers of different dielectric constants, affecting all the frequencies when present. Horizontal and vertical ice layers
(strata) are formed at various depths in the firn primarily from the refreezing of seasonal snow melts. Over time, older ice
layers move downward due to the snow accumulation while new ice layers are formed for subsequent melts at the top layers,
creating a complex set of stratigraphy and significantly influencing the L-band signals from the deeper layers. Therefore, L-
band TB is determined by the subsurface temperature, stratigraphy, and LWA.
**2.2.2 Frozen Season Response**
L-band TB exhibit some distinct spatial features over GrIS during a frozen season. Along a typical west-east transect, TB is
the highest in the ablation zone, then it gradually decreases to its lowest value in the percolation zone, followed by a gradual
increase towards a moderate value in the upper accumulation zone. A mirror image is seen on the eastern side of the ice sheet.
The spatial features of H-pol TB are similar to V-pol TB, but it is more affected by sub-surface layering. This is illustrated in



Figure 1 with V- and H-pol mean frozen season TBs and their normalized polarization ratio (NPR, defined as NPR = (TBV-
TBH)/(TBV+TBH)). The ablation zone is characterized by exposed glacial ice with a high density and internal temperature
than those of the ice sheet towards Greenland's interior. It is soaked and swept by a large amount of meltwater every year.
During the frozen season, the L-band emission has a high effective emissivity, radiating the warmer physical temperature of
the deeper layers. In the percolation zone, on the other hand, moderate, but varying melt occurs almost each or every few years
that percolates down and refreezes at different depths forming discrete ice layers and ice pipes, causing substantial scattering
of mean TB (Jezek et al., 2018). High NPR values highlight the area with dense ice layers (strata). The upper accumulation
zone experiences light or no melt but accumulates snow, resulting in less density variation compared to the percolation zone.
For detecting melt and quantifying LWA, we used vertically polarized TB (V-pol TB) considering its lower sensitivity to sub
surface stratigraphy.

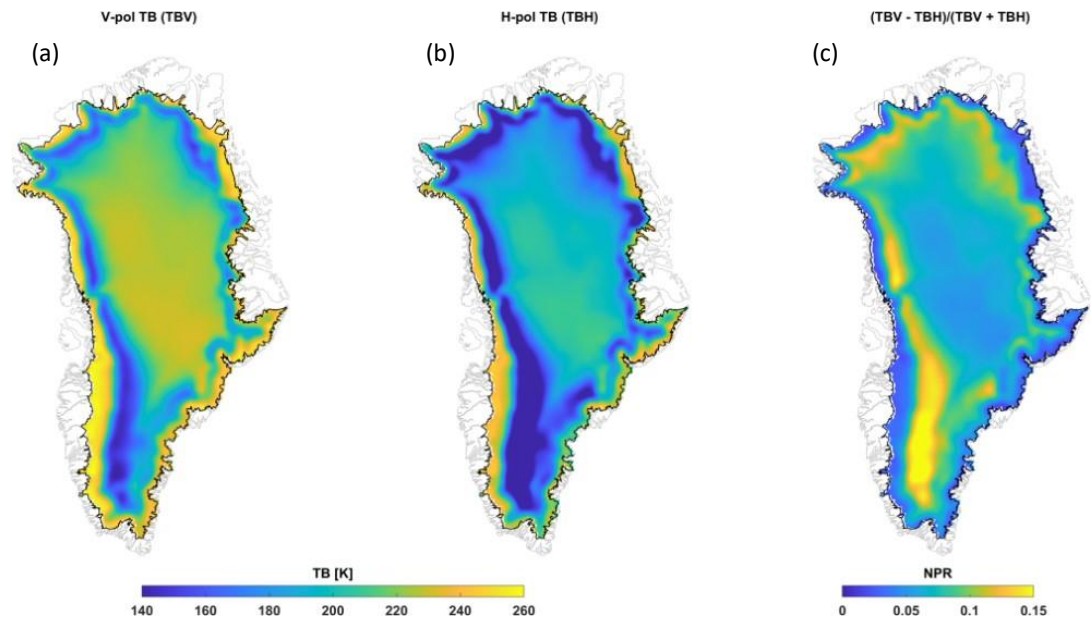


**Figure 1: L-band radiometric response of GrIS during frozen season. Vertically polarized TB (a), horizontally polarized TB (b)**
**averaged over Jan 1 – Apr 7th, 2015 – 2023, and their normalized polarization difference (c).**
**2.2.3 Melt Season Response**
During summer season in presence of melt, the L-band TB generally decreases in the ablation and upper accumulation zones
compared to the frozen season, while it increases significantly in the percolation zone. Fig. 2 illustrates this for a sample
summer day (Jul 31, 2019) when melt was detected across the K-transect (~ 67° constant latitude; see the red line in Figure
2a). The melt flags (square symbols over dashed line) specify the TB samples for which melt was detected (Sec. 3.2). The




presence of LWC in the snow and firn increases the absorption and emission in turn (Mote and Anderson, 1995). However, at
the lower elevation around the ablation zone, the TB decreases from its very high level (~260 K) as the LWC of the seasonal
snow layer increases. This is because, when the LWC in the snow layer exceeds a threshold, snow becomes saturated and it
creates a reflective boundary at the ice and snow interface, suppressing the emission from the ice layer and resulting in overall
lower TB. This is caused by intense melting common in the ablation zone (Figure 2b). The percolation zone experiences
moderate melt, making the snow and firn highly absorptive during melt season. As a result, the TB gradually increases from
its winter references (Figure 2b), making the L-band sensitive to the total amounts of melt. In the upper accumulation zone,
melt seldom occurs. But when it occurs, it may percolate and refreezes quickly in the colder snow creating ice layers that cause
reflection, reducing L-band TB signals.

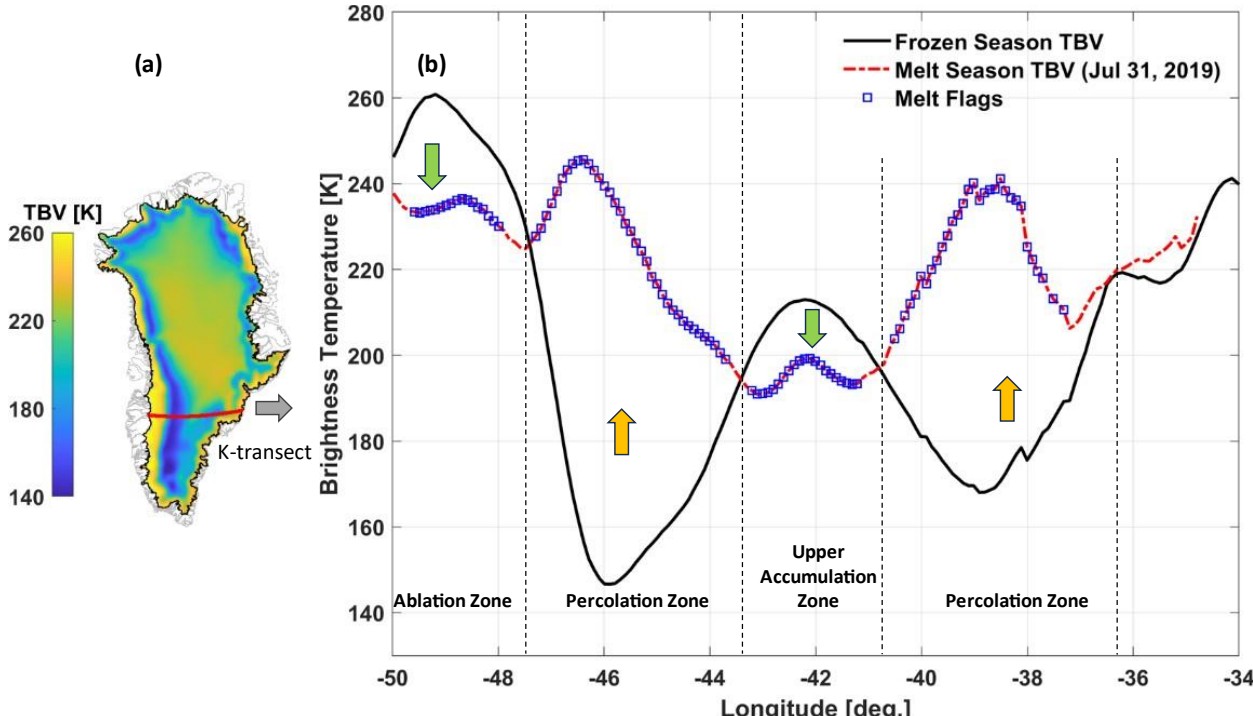


**Figure 2: Radiometric response of L- band TB during frozen and melt season. The location of K-transect is highlighted by red line**
**over the mean frozen season TBV map (a). Corresponding TBs across the transect are shown in (b): the black line represents the**
**mean V-pol TB during the frozen season (Jan 1 – Apr 7th of the same year). The red dash-dotted line indicates TB responses on a**
**sample melt day (Jul 31, 2019). The blue square symbols on the red dashed-dotted line depict melt flags (melt detections).**
**Approximate location of ablation, percolation, and upper accumulation zones are depicted along the K-transect for reference.**
Figure 3 shows the L-band V-pol TB time series during Mar – Oct 2016 at the DY2 AWS, a location representative
of the percolation zone. During the frozen season, the L-band TB is relatively lower and stable. During the melt season, it
captures the diurnal signals during melting phases (melt generation). However, it diminishes as the melt percolates to deeper
layers. From the onset through the end of the melt season, the density and grain size increase in the snow and firn layers due



to melt (Vandecrux et al., 2022). Although the L-band TB is relatively insensitive to the grain growth, the post-melt TB level
may still decrease because of increased reflection from newly formed ice layers. This effect is pervasive, especially across the
accumulation zone justifying a dynamic threshold in threshold-based melt detection algorithms.

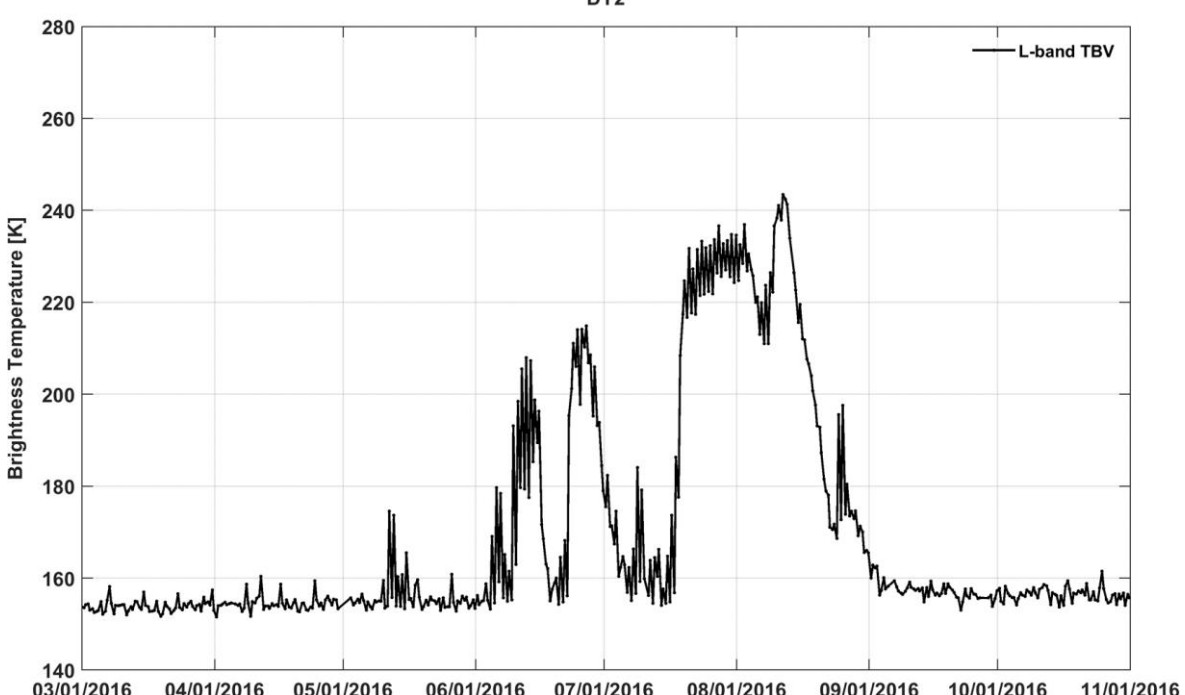

**Figure 3: L- band V-pol TB time series at the DY2 automatic weather station location during** during Mar – Oct, 2016 **illustrating**
**the change of TB level caused by melting, snow accumulation, and other physical processes.**
**2.3 Melt Retrieval Algorithm**
We used a threshold-based empirical detection algorithm to detect surface and subsurface melt events. The threshold is
determined by:
$T = \mu \pm m\sigma$ (4)
where $\mu$ is a reference TB (the mean during the frozen season), σ is the standard deviation of the TB during the reference
period, and m is an empirically derived constant. A constant value of 10 was chosen for $m$. First, to detect the first and last
melt during a year for a grid point, mean TB during Jan 1 – Apr 7[th], and October 24 – December 31 was used as the reference
values, respectively. An averaged value of $\sigma_{spring}$ and $\sigma_{fall}$ is used with final adjustment of $m$ in such a way that the threshold
does not miss the first and last melts. Then a linearly transitional reference value is used between the first and last melt days
to account for the change in TB value for subsequent melt because of refreezing.



We used an inversion of a simplified ice sheet emission model to estimate the LWA and physical properties of the
detected melt events. The retrieval algorithm consists of a multi-layer forward model (Fig. 4) simulating the L-band TB
(Mousavi et al., 2021) and a cost function minimization between the simulated and observed TB. The forward model represents
the ice sheet as a stack of N+1 vertical layers, where each layer is characterized by its complex dielectric constant ($\epsilon$), density
($\rho$), physical temperature ($T$), and thickness ($d$). The top and the bottom layers are assumed to be semi-infinite, while the
intermediate layers are configured with variable thicknesses. The TB is then modelled using the incoherent approach of
radiative transfer (RT) theory, without considering the effects of volume scattering analytically (but considering its dielectric
effects explicitly). For a specific depth z, the upwelling TB, for a given polarization, $p$, is given by:
$$T_{Bnp}^{u}(\theta_n, z) = \left[\Gamma_{np} T_{Bnp}^{d}(\theta_n, - d_n) + (1 - \Gamma_{np})T_{B(n+1)p}^{u}(\theta_{(n+1)}, - d_n)\right]e^{-k_{an}(z+d_n) \sec\theta_n} + (1 - e^{-k_{an}(z+d_n) \sec\theta_n}) T_n$$

(5)

where $T_{Bnp}^{u}$ and $T_{Bnp}^{d}$ represents the upwelling and downwelling p-polarized TB at the interface z = - $d_n$ characterized by
reflectivity $\Gamma_{np}$. $\theta_n$ is the incidence angle determined from the Snell's law and dielectric constant, and $k_{an}$ is the power
absorption coefficient given by $k_{an}$ = - 2Re{ $\omega\sqrt{\varepsilon_n\mu_0}$ }, where ω is the angular frequency, $\varepsilon_n$ is the complex permittivity of
the layer, and $\mu_0$ is the magnetic permeability for a nonmagnetic material. $T_n$ is the physical temperature of the layer and
assumed to be homogenous within the layer. The downwelling part of the TB, $T_{Bnp}^{d}(\theta_n, - d_n)$, is given by:
$$T_{Bnp}^{d}(\theta_n, - d_n) = \Gamma_{(n-1)p} T_{Bnp}^{u}(\theta_n, - d_{n-1}) + (1 - \Gamma_{(n-1)p}) T_{B(n-1)p}^{d}(\theta_{(n-1)}, - d_{n-1})$$    (6)
It is assumed that there are no downward and upward emissions beyond the top (z = 0) and bottom (z = - $d_N$) semi-
infinite layers respectively, and the atmospheric attenuation is also to be negligible considering L-band frequency. Therefore,
the top-of-the atmosphere TB is found from equation (1),
$$T_{BP}(\theta_0, H) \approx T_{B0p}^{u}(\theta_0, 0) = (1 - \Gamma_{0p}) T_{B1p}^{u}(\theta_1, 0)$$    (7)



**Figure 4: L-band multilayer ice sheet forward model.**

Following (Mousavi et al., 2021), the TB simulation considers the percolation zone of the GrIS as a four-layer medium (N=3) during melt season -- air, a wet snow/firn layer, a highly reflective firn layer, and a semi-infinite firn/ice layer (Fig. 4). The highly reflective layer represents the combined reflective effects by the complex stratigraphy due to numerous ice layers that is present in the percolation zone of the GrIS, as well as the effects of volume scattering inside the same layer (Mousavi et al., 2021). It is modeled as a layer of higher complex permittivity. For faster processing during retrieval, we developed separate look-up-tables (LUTs) for dry and melt season prescribing layer parameters by sweeping over a realistic range of each parameter. During dry season, the wet layer is absent (corresponding to dry snow layer that is assumed to be effectively transparent when completely frozen) and the complex permittivity that best matches the measurements is prescribed to the reflective layer.

The LUTs were revised compared to their original versions (Mousavi et al., 2021) in the following way. For each layer, density (varying from a fresh snow density of 50 kg/m$^3$ to that of solid ice of 917 kg/m$^3$) and physical temperature (varying from 200 K to 273.15 K) were prescribed. The dielectric constant of the layers was calculated based on the density and physical temperature following Ulaby and Long, 2014 and Mätzler, 2006. Then, the emission model was run for each combination. The model computes the top-of-the-atmosphere L-band TB at the V- and H-pol based on frequency, incidence angle, and the layers' thickness and dielectric constant. For the melt season, the wet snow layer is inserted with a volume fraction of meltwater, $m_v$, which is varied from 0 to 5% in 40 equally spaced steps, and thickness, $d_{wet}$, which is varied from 10 cm to 20 m, in 10 cm steps for the top 60 cm, 20 cm steps for next 1.4 m, 40 cm steps for next 8 m, and 1 m steps for the



next 10 m. For $m_v > 0$, $T_{wet}$ must be 0 C. With all these constraints, the tuning finally results in two LUTs with six and eight
dimensions for the dry and melt seasons, respectively.
The inversion was performed by optimizing a cost function that minimizes the distance between the LUT-modeled
TB and the corresponding SMAP-measured TB for each 3.125 km grid cell. The optimization was carried out in two steps for
each melting grid. First, the frozen season snow/firn density, physical temperature, and dielectric constant were estimated.
Second, using that information, the volume fraction of meltwater $m_v$, and corresponding wet layer thickness $d_{wet}$, were
determined for a time stamp during the melt season. The LWA is thus the product of the two, i.e., $LWA = m_v\, d_{wet}$ [m] m.w.e.
This represents the instantaneous total LWA present in the SMAP footprint for that time stamp within the SMAP sensing
depth, covering the typical infiltration of the meltwater in the percolation zone as per the climatological records (Samimi et
al., 2020; Vandecrux et al., 2020). The detection algorithm uses both increasing and decreasing summer TBs to generate melt
flags; however, the inversion only considered increasing TBs for LWA quantification. We averaged twice daily LWA outputs
to compute daily samples.

**2.4 Automatic Weather Station Measurements**

Direct measurements of LWA are not available for validation. However, AWS networks, such as the Greenland Climate
Network (GC-Net) (Steffen et al., 1996; Steffen and Box, 2001) or the Programme for Monitoring the Greenland Ice Sheet
(PROMICE) (Fausto et al., 2021), provide essential surface parameters that can be used to estimate LWA with an energy
balance model. The Geological Survey of Denmark and Greenland (GEUS), now manages these two AWS networks, which
cumulate 33 active ice sheet sites in Greenland that provide a suite of measurements, such as incoming/outgoing short and
longwave radiation fluxes, snow-surface height, air temperature, air pressure, vector winds, as well as subsurface temperature
and density profiles (Fausto et al., 2021).
We used the hourly measurements from six PROMICE and GC-Net AWSs in the percolation zone to force an EMB
model that produce a reference LWA, which was then used to validate the LWA retrieved from SMAP observations. The
stations were selected considering their locations (see Fig. 5) and melt climatology. The meteorological forcing governs the
surface energy budget (SEB) and was used to derive a coupled energy balance and snow/firn hydrology model (Ebrahimi and
Marshall, 2016; Samimi et al., 2021) that provide an estimate of hourly LWC evolution within snow and firn.

**2.5 Ice Sheet Energy Balance and Hydrology Model**

The energy balance model (EBM) determines the net energy available for melt by considering the SEB along with modelled
surface temperature, thermal emissivity, and albedo. The coupled model also accounts for the hydrological processes like
meltwater infiltration, refreezing, and retention within the firn. We used two ice sheet EBMs for comparisons with the SMAP
LWA retrievals. A detailed description of these models is out of the scope of this article, but brief descriptions are given below.
Readers are referred to relevant cited articles for further details.



### 2.5.1 Energy Balance and Hydrology Model

A locally calibrated and validated EBM (Ebrahimi and Marshall, 2016; Samimi et al., 2020, 2021) was used as the primary reference for comparison. The EBM was initialized with ice core density profiles, stratigraphy, and the sub-surface temperature profiles (Vandecrux et al., 2023b) and forced with the hourly surface forcing from PROMICE and GC-Net AWS. The model first calculates the net energy balance from the surface forcing by combining the energy fluxes towards the surface layer. Then, it runs a subsurface model to calculate heat conduction and melt rates in the upper 20 m of the snow/firn by resolving the profile into 43 vertical layers, with gradually decreasing thickness near the surface.

When the surface temperature reaches the melting point, and the net energy is positive, melting occurs. Conversely, if net energy is negative and the surface layer is at the melting point, any existing liquid water will freeze, releasing latent heat and causing the surface layer to cool until all liquid water is refrozen, depending on the energy balance. When surface layer temperatures are below the melting point, and there is either an excess or deficit of energy leading to warming or cooling, the energy balance within a one-dimensional model of subsurface temperature evolution determines the subsurface temperature and density profiles. The model determines hydraulic conductivity and permeability after Meyer and Hewitt (2017), while thermal conductivity was modeled following Calonne et al., (2019). The profile then governs the availability of local water at any level for the next time stamp. The model relates to a basic approach to how meltwater flux percolates downward using Darcy's law. The local water balance is determined by mass conservation in each subsurface layer. Once a layer becomes temperate, it can retain liquid water within its pore space or allow it to percolate deeper (Coléou and Lesaffre, 1998). The subsurface model is coupled with a hydrology model that redistributes the meltwater; depending on the subsurface temperature profile, the meltwater may refreeze. Due to refreezing, density may increase, and ice layers may form that may reduce or completely block meltwater infiltration. The firn densification was modeled as in Vionnet et al., (2012). We henceforth refer to this model with AWS forcing as the EBM for simplicity. To evaluate the LWA retrieval, we calculate the daily average LWA from the hourly EBM output.

### 2.5.2 Glacier Energy and Mass Balance (GEMB) Model

We used output from GEMBv1.0 as a secondary source of comparison. It is a module in the Ice-sheet and Sea-level System Model (ISSM, https://issm.jpl.nasa.gov/) that models the ice sheet surface-energy and mass exchange and snow/firn state in a 1D column over time (Gardner et al., 2023). It has more than 100 vertical layers with <5 cm thickness in the top layers and employs spatially variable grid size based on the ice sheet dynamics. GEMB formulates irreducible water content according to Colbeck, (1973), and uses bucket scheme (Steger et al., 2017) for liquid water infiltration. Parameterization of firn densification and thermal conductivity follow Herron and Langway, (1980) and Sturm et al., (1997) respectively. Readers are referred to Gardner et al., (2023) and references therein for further details. The model was forced with 3-hourly ERA5 (Hersbach et al., 2020) atmosphere and radiation conditions, after the methods described by Paolo et al., (2023). For the GrIS output presented here, the ERA5 surface temperature and downwelling longwave radiation forcing were spatially bias-



corrected for each month, such that all values were adjusted by the difference between the RACMO2.3 (Noël et al., 2016) and
the ERA5 1980-2015 monthly means. GEMB outputs were provided on the ISSM native grid, daily from 2015 through 2023
and included temperature, density, and LWC profiles. The daily output of the closest grids to the selected AWSs was used.
We henceforth refer to this model as GEMB.

### 2.5.3 Evaluation Metrics

To compare SMAP LWA time series with EBM and GEMB model, we considered the standard evaluation metrics including
mean difference, standard deviation (STD), mean absolute difference (MAD), Pearson linear correlation coefficient (r), root
mean square error (RMSE). We also compared day of melt onset (the first day of summer melt) and melt freeze up (the last
day of summer melt), summer melt duration (difference of melt onset and freeze up), maximum summer LWA, and annual
sum of daily LWA (LWA$_{YS}$). To determine the day of melt onset and freeze up, we only considered melt events with LWA >
2 mm, to avoid any spurious melts that may result from any instrumental noise or other sources. The LWA$_{YS}$ is the sum of
daily LWA over a year. It is a measure of the total seasonal LWA, but it does not represent the total surface melt generated
over a year. This is because SMAP observes the instantaneous LWA, the net water balance, which is the cumulative sum of
surface melt, refreezing, and runoff over SMAP footprint. When the net water balance remains positive overnight, it can be
considered multiple times in the total integrated LWA as long as it persists.

### 3 Results

### 3.1 Liquid Water Amount

### 3.1.1 Comparison to Locally Calibrated EBM

Figure 5 shows a comparison of the SMAP-retrieved LWA with the LWA derived from the EBM at six different PROMICE
and GC-Net AWS sites for one year. The melt season at CP1 site (Figure 5a) began at the fourth week of June according to
both, SMAP and the EBM, and continued through the first week of September according to SMAP, while it extended through
the end of September in the model estimate. Shortly after complete refreezing of the first melt event in late June, SMAP
resumed recording LWA in first week of July. Both SMAP and the EBM closely agree in both phase and magnitude of LWA
during first half of July. Afterwards, the EBM reports overall higher LWA for the rest of the season and it seemed to retain
liquid water for an elongated period when SMAP showed a fully refrozen firn. The overall agreement is given by the Pearson
linear correlation coefficient (r) of 0.82 and root mean square difference (RMSD) of 13 mm. The onset of melt event at KAN_U
site (Figure 5b) is concurrent to CP1 in accordance with the EBM. However, SMAP did not record melt at this site until the
first week of July. Unlike CP1 site, SMAP reports persistent LWA through the first week of October, whereas EBM shows
complete refreezing by the second week of September. Both SMAP and the EBM captured less LWA at KAN_U site compared
to CP1. This is somewhat counter-intuitive because the KAN_U site is located at a lower elevation than CP1 site (see the





elevation in Figure 5d). In fact, KAN_U is characterized by having a lower accumulation and higher melt rate every year
(MacFerrin et al., 2019; Machguth et al., 2016). However, excessive melt have also created thick ice slabs in this location
(MacFerrin et al., 2019; Machguth et al., 2016). As a result, liquid water cannot percolate to the deeper layers and run off
horizontally. The model excludes this liquid water in the form of 'drainage', and SMAP only sees the existing meltwater in its
field of view. At the DY2 site (Figure 5c), LWA estimated by SMAP, and the EBM resemble more closely both in phase and
magnitude (except the difference in timing of complete freeze up). This is reflected by nearly perfect correlation and a small
overall RMSD (3 mm) as shown.
SMAP LWA also closely aligns with the EBM at NSE site in magnitude and duration of liquid water presence (r =
0.98 and RMSD = 2 mm) although SMAP seemed to miss the late August small melt event (Figure 5e). The agreement however
exhibits the greatest deficiencies at SDL site for this melt season (Figure 5f). Although the timing of the melt onset and late
August secondary melt event matches precisely, the EBM reports overall a higher LWA and an extended summer melt duration
at this location. This is manifested in the performance metrics shown by a relatively higher RMSD (23 mm) and lower
correlation coefficient (0.77). The performance at SDM site is generally good (r = 0.92, and RMSD = 6 mm), except the EBM
demonstrates a delayed refreezing than SMAP (Figure 5g).
It is pertinent to highlight that while *in situ* LWA at all these AWS were derived from the energy balance model
forced by the pointwise measurements at the AWS locations, the SMAP retrievals estimated a spatially averaged LWA
corresponding to the ~30 km effective resolution of the enhanced-resolution TB. Approximately, during the first half of the
melt season, the LWA is primarily determined by meltwater generation in response to the net radiation flux at the surface.
Whereas, roughly during the second half when the net radiation flux remains negative, refreezing becomes the dominant
process. Hence, the model's representation of the surface melt infiltration, heat transfer, and other physical processes play a
significant role, posing additional uncertainties. The AWS measurements that run the model also add some inherent
uncertainties. Therefore, assessing relative accuracies is not straightforward. Nevertheless, the general agreements between
the model and SMAP retrieved LWA in magnitude and phase at these locations suggest that the spatial heterogeneity of melt
processes is not acute in these areas.



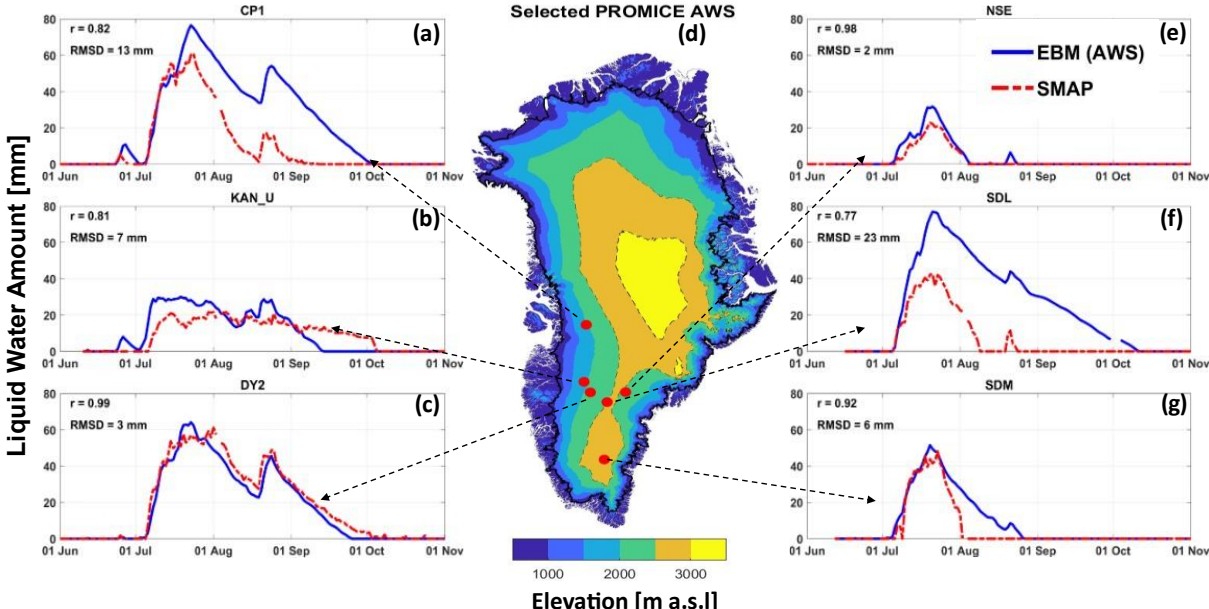

**Figure 5: Comparison of the total daily liquid water amount retrieved from SMAP (red dashed lines labeled SMAP) and estimated by the EBM forced with *in situ* measurements (blue lines labeled EBM) at selected PROMICE and GC-Net AWS within the GrIS percolation area. The locations of the AWSs are shown in the middle panel along with the ice sheet surface elevation (Howat et al., 2014).**

**3.1.2 Three Way Comparison: SMAP, EBM, and GEMB Model**

We performed a pairwise comparison among SMAP, EBM, and GEMB models (Figure 6) for the 2021, 2022 and 2023 melt seasons (based on available meteorological data) at the 6 AWS locations. In detail performance metrics are documented in Table 1 (mean difference, STD, mean absolute difference, Pearson linear correlation coefficient, and RMSD) and Table 2 (melt onset, freeze up, duration of summer melt, maximum summer melt, and annual sum of daily LWA). Because of SMAP outage for 2022 summer, performance metrics in Table 1 only considered the operational part of SMAP. Table 2, however, excludes SMAP for 2022 melt season except the melt onset information as the other metrics were impacted by the outage.

At KAN_U site, the overall agreement between SMAP and the EBM was determined to be better (r > 0.8) than the agreement either SMAP and GEMB model (r < 0.7), or the EBM and GEMB model (r < 0.5) for the 2021 and 2023 melt seasons (Figure 6a-6c). All the AWS data required to run the EBM for 2022 melt season were not available. GEMB model demonstrates earlier melt onset and freeze up and thus shortest summer melt duration in 2021 and 2023 melt seasons compared to both SMAP and the EBM. SMAP estimated a maximum summer melt of 56 mm at this site in 2021 melt season, while both the EBM and GEMB model recorded maximum summer melt of 34 mm in 2023 melt season. No pair shows consistent superiority at CP1 site






**Figure 6: Comparison of the SMAP retrieved total daily liquid water amount (red dashed lines) with the estimated LWA from EBM**

**(blue solid lines), and GEMB model (black dotted lines) at selected PROMICE and GC-Net AWS within the GrIS percolation area.**





(Figure 6d-6f). SMAP LWA generally aligns closer with GEMB model LWA in 2021 melt season, and with the EBM in 2022
melt season, whereas in 2023 melt season, the EBM and GEMB model matches closer to each other than to SMAP. DY2 lacks
AWS forcing during 2021 melt season. So, EBM result is missing for this melt season. Between SMAP and GEMB model,
the later estimates overall more LWA ($LWA_{YS}$ 1196 mm vs 1708 mm). There is a close alignment between the peaks of the
two LWA time series, one during the first week of August when SMAP reports about 30 mm LWA, while GEMB model
indicates no melt (Figure 6g). The overall RMSD was found to be 7.24 mm. For the other two melt seasons (Figure 6h-6i),
SMAP and the EBM results show superior agreements (r > 0.96 and RMSD ~ 3 mm). GEMB model reports significantly lower
LWA, both in magnitude and duration (only 564 mm $LWA_{YS}$ compared to 2893 mm (SMAP) and 2608 mm (EBM) resulting
in a higher overall RMSD (>15 mm) with the other two.

**Table 1: Pairwise performance comparison among (a) SMAP, (b) EBM, and (c) GEMB models. Cells are left blank for missing data.**

| | | Mean Difference (mm) | | | Standard Deviation (STD) (mm) | | | Mean Absolute Difference (MAD) (mm) | | | Pearson Correlation Coefficient (r) | | | Root Mean Square Difference (RMSE) (mm) | | |
|---|---|---|---|---|---|---|---|---|---|---|---|---|---|---|---|---|
| AWS | Year | a-b | a-c | b-c | a-b | a-c | b-c | a-b | a-c | b-c | a-b | a-c | b-c | a-b | a-c | b-c |
| KAN_U | 2021 | -3.20 | -4.70 | 0.25 | 8.94 | 14.31 | 6.22 | 3.53 | 6.34 | 2.37 | 0.89 | 0.20 | 0.36 | 9.48 | 15.04 | 6.22 |
| | 2022 | | 0.17 | | | 5.98 | | | 2.08 | | | 0.71 | | | 5.98 | |
| | 2023 | 1.08 | -2.60 | 5.07 | 6.72 | 7.58 | 9.48 | 4.55 | 4.10 | 5.90 | 0.81 | 0.26 | 0.49 | 6.79 | 8.00 | 10.73 |
| CP1 | 2021 | 8.65 | 1.82 | 6.50 | 10.58 | 7.45 | 13.30 | 8.65 | 4.06 | 8.98 | 0.88 | 0.74 | 0.65 | 13.64 | 7.64 | 14.77 |
| | 2022 | 0.19 | 0.81 | -1.08 | 0.71 | 3.49 | 3.58 | 0.24 | 0.82 | 1.12 | 0.97 | 0.87 | 0.87 | 0.74 | 3.58 | 3.73 |
| | 2023 | 5.82 | 7.50 | -1.59 | 11.74 | 12.92 | 5.63 | 6.07 | 7.52 | 1.83 | 0.82 | 0.89 | 0.97 | 13.09 | 14.93 | 5.84 |
| DY2 | 2021 | | 1.26 | | | 7.17 | | | 2.73 | | | 0.75 | | | 7.27 | |
| | 2022 | 1.48 | 0.74 | 1.64 | 2.87 | 5.66 | 6.55 | 1.51 | 2.90 | 3.54 | 0.96 | 0.37 | 0.53 | 3.22 | 5.68 | 6.73 |
| | 2023 | -1.03 | -7.59 | 5.87 | 2.76 | 15.72 | 14.29 | 1.40 | 8.08 | 6.45 | 0.99 | 0.42 | 0.41 | 2.94 | 17.44 | 15.43 |
| SDM | 2021 | 3.75 | 2.43 | 1.02 | 7.19 | 5.87 | 6.51 | 3.78 | 2.56 | 3.46 | 0.61 | 0.63 | 0.70 | 8.09 | 6.34 | 6.57 |
| | 2022 | 0.09 | 0.08 | 0.14 | 0.31 | 0.64 | 0.73 | 0.09 | 0.08 | 0.19 | | | 0.31 | 0.32 | 0.64 | 0.75 |
| | 2023 | 2.56 | 0.32 | 1.58 | 5.34 | 2.37 | 4.29 | 2.70 | 0.62 | 2.10 | 0.92 | 0.97 | 0.94 | 5.91 | 2.38 | 4.56 |
| SDL | 2021 | 4.83 | 2.02 | 2.49 | 7.92 | 5.23 | 8.57 | 4.83 | 2.13 | 4.93 | 0.70 | 0.71 | 0.53 | 9.25 | 5.59 | 8.91 |
| | 2022 | 0.23 | 0.23 | -0.10 | 1.03 | 1.32 | 1.46 | 0.23 | 0.23 | 0.35 | 0.70 | 0.30 | 0.44 | 1.05 | 1.34 | 1.46 |
| | 2023 | 15.73 | 3.87 | 6.03 | 16.46 | 8.81 | 10.95 | 15.76 | 3.87 | 7.76 | 0.77 | 0.95 | 0.88 | 22.73 | 9.61 | 12.48 |
| NSE | 2021 | 1.45 | 1.97 | -0.52 | 3.85 | 5.04 | 3.77 | 1.46 | 1.98 | 1.54 | 0.16 | 0.27 | 0.66 | 4.10 | 5.40 | 3.79 |
| | 2022 | | 0.07 | | | 0.47 | | | 0.07 | | | | | | 0.48 | |
| | 2023 | 0.91 | 2.97 | -3.65 | 2.24 | 8.20 | 8.05 | 0.91 | 2.97 | 3.76 | 0.98 | 0.94 | 0.94 | 2.41 | 8.70 | 8.82 |

As per maximum summer melt and $LWA_{YS}$, SDL, SDM and NSE sites experienced the highest LWA in 2023 melt
season compared to the other two melt seasons under consideration (Figure 6j-6r). SMAP did not record any LWA in any of
these sites during 2022 melt season when EBM (except NSE where AWS data were not available), and GEMB models also





reported the least LWA in three melt seasons (Figure 6k, 6n, and 6q). In 2021 melt season, SMAP estimated overall lower
LWA and shorter summer melt duration than that of EBM and GEMB models in these sites. But the agreements between EBM
and GEMB models are in the same orders (see Table 1 and Table 2), with EBM exhibited delayed refreezing consistently.
**Table 2: Comparison of individual performances: (a) SMAP, (b) EBM, and (c) GEMB models. A threshold of 2 mm**
**LWA was considered to avoid any spurious melt event. Cells are left blank for missing data.**

| | | Melt Onset (DOY) | | | Melt Freeze up (DOY) | | | Summer Melt Duration (days) | | | Maximum Summer LWA (mm) | | | Annual Sum of Daily LWA (mm-year) | | |
|---|---|---|---|---|---|---|---|---|---|---|---|---|---|---|---|---|
| **AWS** | **Year** | SMAP | EBM | GEMB | SMAP | EBM | GEMB | SMAP | EBM | GEMB | SMAP | EBM | GEMB | SMAP | EBM | GEMB |
| **KAN_U** | 2021 | 179 | 175 | 173 | 254 | 248 | 229 | 76 | 74 | 57 | 56 | 24 | 34 | 2212 | 736 | 652 |
| | 2022 | 186 | | 168 | | | 270 | | | 103 | | | 33 | | | 1004 |
| | 2023 | 188 | 176 | 163 | 277 | 254 | 240 | 90 | 79 | 78 | 22 | 30 | 34 | 1302 | 1504 | 459 |
| **CP1** | 2021 | 179 | 175 | 173 | 251 | 277 | 247 | 73 | 103 | 75 | 39 | 61 | 47 | 842 | 2471 | 1191 |
| | 2022 | 181 | 181 | 180 | | 252 | 254 | | 72 | 75 | | 16 | 31 | | 175 | 435 |
| | 2023 | 175 | 176 | 175 | 246 | 273 | 277 | 72 | 98 | 103 | 61 | 76 | 91 | 1569 | 3458 | 4029 |
| **DY2** | 2021 | 179 | | 173 | 251 | | 252 | 73 | | 80 | 35 | | 54 | 1196 | | 1708 |
| | 2022 | 185 | 185 | 169 | | 270 | 269 | | 86 | 101 | | 30 | 29 | | 838 | 513 |
| | 2023 | 187 | 187 | 151 | 313 | 265 | 242 | 127 | 79 | 92 | 61 | 64 | 35 | 2893 | 2608 | 465 |
| **SDM** | 2021 | 200 | 175 | 174 | 232 | 256 | 241 | 33 | 82 | 68 | 35 | 40 | 38 | 164 | 857 | 663 |
| | 2022 | | 245 | 169 | | 269 | 171 | | 25 | 3 | | 8 | 9 | | 32 | 25 |
| | 2023 | 188 | 187 | 186 | 213 | 237 | 220 | 26 | 51 | 35 | 48 | 52 | 50 | 775 | 1202 | 890 |
| **SDL** | 2021 | 200 | 185 | 174 | 235 | 261 | 239 | 36 | 77 | 66 | 21 | 41 | 35 | 195 | 1096 | 607 |
| | 2022 | | 169 | 169 | | 248 | 248 | | 80 | 80 | | 7 | 14 | | 68 | 95 |
| | 2023 | 187 | 187 | 176 | 235 | 282 | 246 | 49 | 96 | 71 | 43 | 77 | 77 | 919 | 3383 | 2196 |
| **NSE** | 2021 | 201 | 185 | 177 | 201 | 241 | 235 | 1 | 57 | 59 | 3 | 19 | 29 | 5 | 271 | 371 |
| | 2022 | | | 171 | | | 197 | | | 27 | | | 5 | | | 21 |
| | 2023 | 188 | 187 | 176 | 214 | 234 | 232 | 27 | 48 | 57 | 23 | 32 | 61 | 338 | 508 | 1314 |


### 3.1.3 SMAP LWA Time Series

Figure 7 shows the SMAP retrieved LWA time series at the mentioned six AWS locations on the southwest and southeast
sides of the GrIS percolation zone. The time series do not include results during 2019 and 2022 outage. As evidenced, the
AWS sites in the southwest sites (Figure 7a-7c) experienced more average LWA and longer summer melt duration than the
AWS sites in southeast sites (Figure 7d-7f). SDL, SDM, and NSE witnessed an insignificant LWA (<10 mm) during 2015 -
2020 melt seasons. However, it was found to be increasing in recent years (Figure 7d-7f). SMAP recorded the highest LWA





in 2023 melt season during 2025 - 2023 at all the AWS locations, except at KAN_U where 2021 marked the highest melt
season.

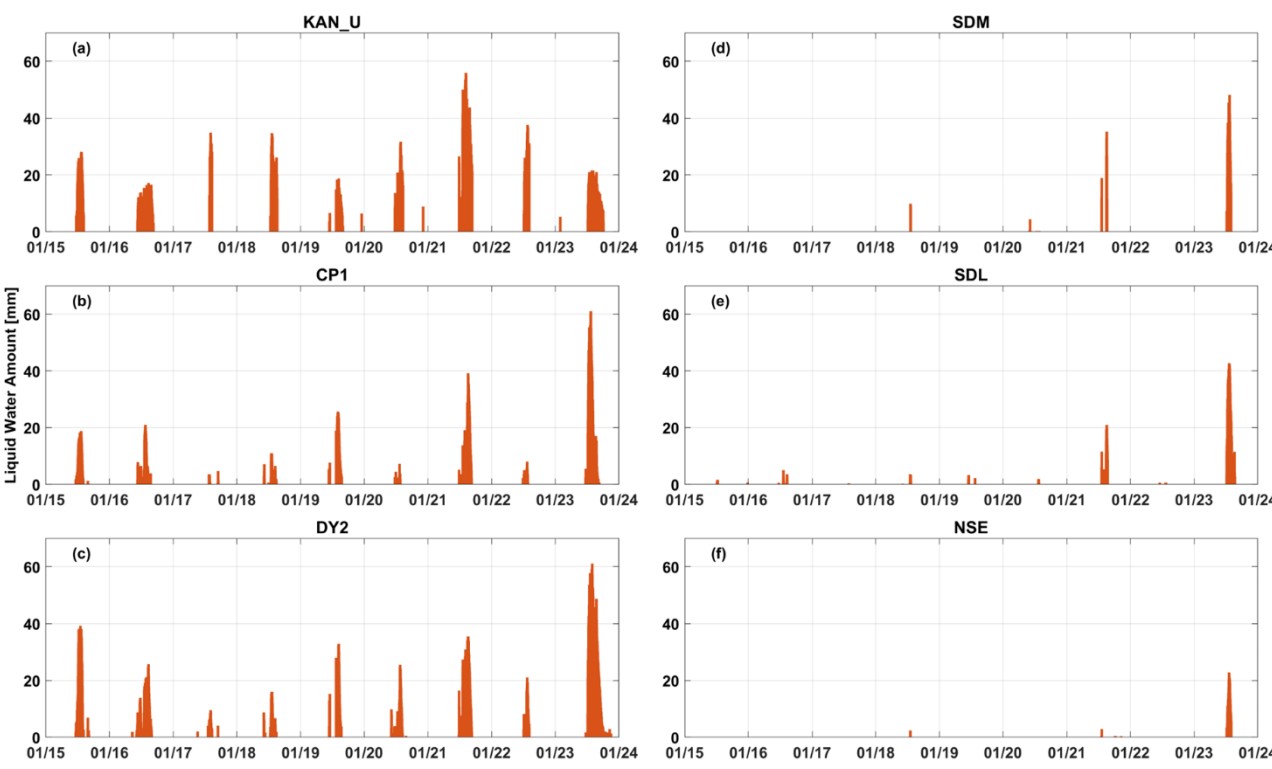


**Figure 7: SMAP retrieved LWA time series for 2015-2023 period at at six selected PROMICE and GC-Net AWS within the GrIS**
**percolation area.**
**3.1.4 Spatial Variability**
Figure 8 illustrates the annual sum of daily LWA (LWA$_{YS}$) for 2015 – 2023. Here, we masked the area where melt is detected
by decreasing summer TBs (compared to winter reference). As mentioned in Sec. 2.2.3, current LWA quantification algorithm
applies to increasing TBs only. This excluded the melt flags in the ablation zone and upper accumulation zone as indicated by
grey shades in Figure 8. There were also some occasions when summer TB decreases below the winter threshold in the
percolation too. Those anomalies were probably caused by short lived melt events that refroze between SMAP passes and
impacted TBs. These anomalies are also masked and not included in the results. As depicted, SMAP captured the similar
spatial trends of LWA distribution across the percolation zone of GrIS as reported by previous studies (Van Den Broeke et al.,
2016; Houtz et al., 2021). In the time frame under consideration, 2023 melt season (Figure 8i) had the highest LWA$_{YS}$ (2634
mm on average for the percolation area) while 2017 (Figure 8c) had the lowest value (757 mm on average for the percolation
area). In 2015 (Figure 8a), the northern ice sheet exhibited a relatively high LWA$_{YS}$; similar intensity and extent were also





recorded for 2023 (Figure 8i). Notably, the melt extended to upper elevations in the dry snow zone in 2021 and 2023.
Unfortunately, SMAP outages in 2019 (Figure 8e) and 2022 (Figure 8h), lead to incomplete coverage for those years and lower
LWA$_{YS}$. It is worthwhile to reiterate that the integrated LWA is a measure of the total seasonal LWA in the specified area.

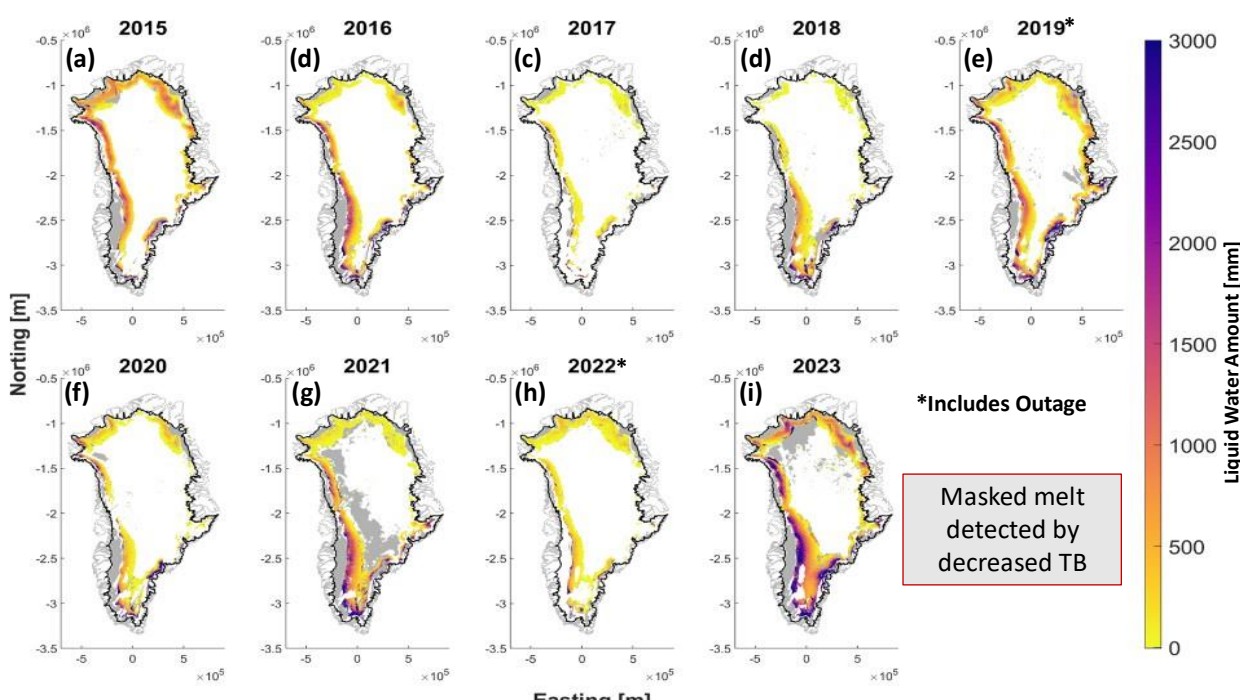

**Figure 8: Total annual sum of SMAP daily LWA for 2015 - 2023. The black solid line represents GrIS edges, and the grey color masks inside the ice sheet indicate melt detections by decreasing TB, which were not quantified.**

## 4 Discussion

The L-band radiometry has the unique advantage of receiving the emission from the deep layers of ice sheets, offering the
opportunity to track meltwater from deeper layers. We have demonstrated its capability to estimate the seasonal LWA that
generally agrees with two state-of-the-art ice sheet models, one locally calibrated and forced with independent *in situ* AWS
measurements and the other forced with ERA5 reanalysis. The legitimacy of spatial and temporal variability shown in SMAP
retrieval for the percolation area of the GrIS is promising.

There are some disagreements as well, but those do not necessarily indicate a deficiency of the SMAP retrievals since

both the references are models with their own limitations. The differences between model results and SMAP retrievals are not
systematic, so they are difficult to explain; but there is no evidence of a consistent bias. Nonetheless, some of the discrepancies
between these estimations of LWA stem from the scale at which those datasets operate. The SMAP LWA was estimated from
the TB measurements averaged over a large footprint and a short integration time. Further, rSIR enhanced-resolution data





products involve overlapping observations to produce the 3.125 km gridded data but still has an effective spatial resolution of
~30 km. Thus, it represents near-instantaneous vertically integrated LWA, averaged over the grid point, whereas the AWS
data is the hourly average of 'point' measurements representative of the 0.1-1 km surrounding the station. The total LWA from
AWS-forced data is the hourly-averaged, vertically integrated net water balance which is determined as the cumulative sum
of hourly surface melt generation, refreezing, and drainage. The surface melt generation is driven by the net surface energy
balance (net radiation and turbulent heat fluxes), which involves uncertainties (e.g., the surface albedo and roughness; errors
in the meteorological inputs), while how the melt and heat are distributed in subsurface firn involves additional uncertainties,
including sensitivity to initial conditions (e.g., the firn temperature and density profile; Samimi et al., 2020). These models
transform surface meteorological information into an amount of surface melt relying on loosely constrained parameterizations
(Covi et al., 2023). Eventually, the models' formulation for the meltwater infiltration is still poorly constrained (e.g. Vandecrux
et al., 2020). Additionally, both the models we used (like other state-of-the-art firn models) are one dimensional – they only
consider vertical movement of water and heat and do not account for horizontal advection. However, firn hydrological
processes are complex and heterogeneous, and processes such as ice layer formation are intrinsically three-dimensional. What
the models consider as 'drainage' (meltwater that moves out of the system) both vertically and horizontally could still be within
the SMAP sensing depth and horizontal footprints. Hence, the comparison should be considered accordingly.

One important disagreement between SMAP and EBM LWA estimation, especially during the refreezing periods, is

that EBM retained LWA for an extended period when SMAP showed complete refrozen condition (Figure 6). We used SUMup
subsurface temperature measurements (Vandecrux et al., 2023b) to verify the cases for which SUMup data are available. One
example is shown in Figure 9. It compares the model-estimated subsurface temperature (Figure 9a) corresponding to the 2021
LWA at CP1 (Figure 6d) to the *in situ* measured subsurface temperature (Figure 9b). It is evident that although the penetration
depth of the model wetting front closely matches the observation, the measurement demonstrates a higher and faster refreezing
compared to the model. The subsurface measurement shows a fully refrozen condition by early September (closely agreeing
with what was revealed by SMAP, see Figure 6d). However, the model seems to retain the subsurface meltwater with a
persistent wetting front even past the beginning of October, which seems unlikely. Speculating extra melt production due to
possible error in the AWS surface forcing, and other surface processes in the EBM, we examined modelled subsurface
temperature profile by reducing surface melt with different factors (<1). We also performed similar analysis with irreducible
water content, thermal conductivity. In either case (not shown), we could not match the subsurface profile with measured
profile within reasonable agreements. This incites questions regarding the model representation of meltwater infiltration, heat
transfer, and refreezing.

The models do not include meltwater infiltration by finger flow (piping). Some recent studies have shown that this is

an important mechanism for moving liquid water from the surface to deep depths (e.g., Vandecrux et al, 2020). The piping
events are short-lived penetration and refreezing events. SMAP will measure the LWC in the piping event, even when it passes
the wetting front, unless the water is refrozen before the SMAP measurement (as is the case with all short-lived melt events).
The model would calculate a certain amount of meltwater based on the surface energy balance, and it would put all this water





into the wetting front layer. However, from the literature and as confirmed by the subsurface temperature measurements (Figure
9b), some fraction of this water would be partitioned into deep piping. The model only includes top-down migration of a
wetting front. This may explain why there are discrepancies between the modelled subsurface temperature profile and the
observed subsurface temperature profile in some cases. Indeed, the deep penetration events causing warming spikes beyond
the wetting front distort the temperature profiles. Therefore, some differences between SMAP and the EBM and GEMB model
could be attributed to this weakness in process representation in the model. But overall, these problems are multifaceted and
additional works are required to understand the basis for these discrepancies. Yet, to this day, there is no observational dataset
that allows to evaluate directly the LWA amount retrieved from satellite observations or calculated by a snow and firn model.

Besides the coarser spatial resolution, SMAP algorithm has its own shortcomings. The emission model simulated
TBs with a simplified view of the stratigraphy which lacks detailed representation of snow and firn microstructures. The model
also neglected atmospheric contributions and assumed homogenous medium and smooth surface within each layer. Although
these effects are not significant at L-band, a detailed characterization was not done. Among other limitations, the detection
algorithm follows a threshold-based technique that uses winter reference of the TB to detect melt events. Aa a result, it is
capable quantifying the seasonal LWA only, not the LWA in perennial firn aquifers which stores a large quantity of the LWA
on the GrIS (Miller et al., 2022a, b). Current algorithm also excluded areas where TB decreases during summer melt. Future
work will be continued to overcome these limitations and refine the algorithm.

To extend the algorithm for GrIS-wide LWA quantification, the ablation zone presents a major challenge. Although
SMAP can detect melt events in the ablation zone, the quantification is difficult for several reasons. The hydrological features
of the ablation zone are markedly different from the percolation or upper accumulation zone. There are widespread networks
of many supraglacial lakes and rivers, crevasses, and other complex heterogeneous factors, such as surface topography, dust
deposition, slush saturation, etc. (Cooper and Smith, 2019; Poinar and C. Andrews, 2021; Smith et al., 2017). This generates
an intricate radiometric response. The average LWA in ablation zone is also significantly higher limiting the L-band emission
in the upper layer only. Houtz et al., (2019) used L-band observations from SMOS satellite to derive LWA at the Swiss Camp
GC-Net AWS located in the ablation zone with a simplistic assumption of fixed (10 cm) wet layer thickness. More *in situ*
observations are needed to characterize the spatial and temporal variability of LWA in the ablation zone.





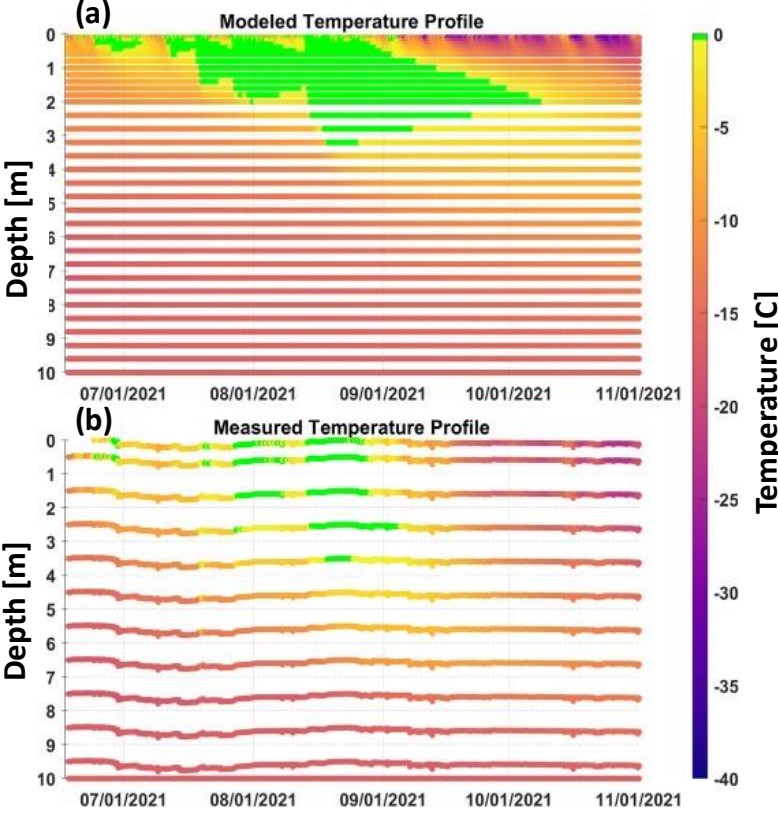

**Figure 9: Modeled (a) and measured (b) subsurface temperatures corresponding to total LWA at CP1 site during 2021 melt season. The 0°C isotherm is highlighted by green color.**

## 5 Conclusion

we have demonstrated quantification of the total surface and subsurface meltwater amount over the Greenland ice sheet using the L-band radiometric observations from the SMAP mission. The retrieval algorithm was described, and the validation results with six *in situ* weather station measurements and reanalysis data were provided. The comparison results were analysed, showing that the retrieval generally agrees with the AWS-driven LWA across the percolation zone. The model uncertainties in representing firn hydrological and thermal processes were explored, and the greatest differences involve the timescale for internal refreezing. The model results commonly predict a longer season for liquid water content in the snow and near-surface firn, i.e., delays in refreezing relative to the SMAP data. Limitations of the SAMP and model estimates LWA, and possible reasons for the discrepancies between them were discussed. Further work is required to understand the basis for these discrepancies. The results demonstrate the potential for providing an observational dataset at time and space scales that will advance our understanding of ice sheet physical processes, helping to better project Greenland's contribution to global sea level rise in response to climate change and variability.



525

**Data and code availability**

SMAP Radiometer Twice-Daily rSIR-Enhanced EASE-Grid 2.0 Brightness Temperatures, Version 2 data products were provided by National Snow and Ice Data Center and are publicly available at https://nsidc.org/data/nsidc-0738/versions/2. The PROMICE hourly AWS measurements are available at https://doi.org/10.22008/FK2/IW73UU (How et al., 2022). The 2023 version of the SUMup subsurface temperature and density profiles are available at https://arcticdata.io/catalog/view/doi:10.18739/A2M61BR5M. SMAP and model LWA are available in a Zenodo repository at https://doi.org/10.5281/zenodo.13800047. The scripts used to perform the analysis for this study can be found at https://github.com/HossanAlamgir/SMAP_MWA_Retrieval_and_Validation_GrIS. MATLAB source code for glacier surface energy balance coupled with firn thermodynamic and hydrological modelling is available in PRISM Data: University of Calgary's Data Repository at https://doi.org/10.5683/SP2/WRWJAZ (Marshall, 2021).

**Author contributions**

AH and AC designed the study and the methodology. AH performed the formal analysis and visualization. NS run the GEMB model and provided the outputs. AH and AC prepared the first draft of the paper. All the authors discussed the results and reviewed the paper. AC supervised the project and managed funding.

**Competing interests**

The authors declare that they have no conflicts of interest.

**Acknowledgements**

This work was funded by the NASA Cryospheric Sciences Program; the work was conducted at the Jet Propulsion Laboratory, California Institute of Technology, under a contract with the National Aeronautics and Space Administration. We gratefully acknowledge computational resources and support from the NASA Advanced Supercomputing Division. The Greenland maps were generated with the assistance of the Arctic Mapping Tools (Greene et al., 2017).

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
