# Peer review of "Retrieval and Validation of Total Seasonal Liquid Water Amounts in"

_EGUsphere, 2024_

## Referee Comment (RC2)

Review comments

general comments

This work continues the author's previous research on the liquid water over Greenland. A radiative transfer model is implemented to generate a Lookup table to retrieve liquid water content during the melt season, and the retrieved results are compared with modeling results. The retrieved LWA shows a good agreement with the modeled results. This paper is of good quality, but I would like to ask a few questions before publication.

Specific comments

1. SMAP data sets. The authors chose the high-resolution SMAP data. What is the benefit of this data set in this research? In the discussion section, could the authors also elaborate more on the sentence"… overlapping observations to produce the 3.125 km gridded data but still has an effective spatial resolution of ~30 km"? How would this affect the results?
2. Permittivity of the wet firn. The author mentioned equation (3), and it seems the authors are using this equation to calculate penetration depth and absorption coefficients in the radiative transfer equation. Could the author discuss the equation used for calculating the effective permittivity?
3. Penetration depth of the wet firn. What would be a typical number and range of the penetration depth?
4. Layering model for the wet firn. Usually, the density profile in the percolation zone increases much faster than in the accumulation zone; the density can be close to the ice after a few meters, say 10m. I'm wondering how the authors deal with the varying density profile in the radiative transfer modeling, given that the penetration depth of 1% wet snow can be 10 meters based on the code provided in Microwave radar and radiometric remote sensing (by Ulaby and Long)?
5. Melt onset. The paper compares the melt onset with the modeling results. Could the auto weather stations provide any information (such as temperature) that can provide some ground truth evidence on the melt onset?
6. Better correlation between SMAP and EBM. In Table 1, the comparison between SMAP and EBM seems to be in agreement. Why is this so? Is this because EBM is using AWS data for calibration? If so, can GEMB somehow be calibrated by the AWS?
7. Possible overestimation of LWA by models. Figure 9 shows the comparison of measured and modeled temperature profiles. The modeled results seem to overestimate the temperatures and, thus, possibly the LWA. Any ideas on resolving this issue and validating the solution?

Thank you.

---

## Author Comment (AC1)

**Author Response to Reviewer #1** (Author responses are highlighted in orange)

This manuscript introduces a highly relevant and impactful application of L-band radiometry in a relatively unexplored research field. The manuscript is generally of very high quality, well written, and with excellent figures. I recommend a few clarifications and comments below. I look forward to seeing the progress on this manuscript and the further development of this method (data product?) in the future.

The authors would like to thank the reviewer for the prompt and comprehensive review and thoughtful comments.

1. At the end of the introduction, you mention and cite a few previous attempts using passive microwave to quantify liquid water on the ice sheet. Can you briefly summarize the work to date and mention how the content of the manuscript adds to or differs from the current status of the subject?

In my opinion one of the primary novelties of the manuscript is the demonstration of liquid water retrieval with single-angle L-band satellite observations. You might also want to consider citing the below paper which also demonstrated single-incidence angle retrievals of snow liquid water but using a ground-based radiometer. *Naderpour, R., Houtz, D., & Schwank, M. (2021). Snow wetness retrieved from close-range L-band radiometry in the western Greenland ablation zone. Journal of Glaciology, 67(261), 27-38.*

We will revise the relevant section in the introduction to include the literature review given above, as follows:

Houtz et al. (2019) used the Soil Moisture and Ocean Salinity (SMOS) multi-angle L-band radiometric observations with a two-layer configuration of the L-band specific MEMLS (LS-MEMLS) model in an inversion-based retrieval framework for simultaneous estimation of snow liquid water content and density at the Swiss Camp site located in the ablation zone of the western Greenland ice sheet (GrIS). This initial study evaluated the results with in situ air temperature and another satellite-based empirical melt detection algorithm (the cross-polarized gradient ratio of 19 GHz and 37 GHz TBs); however, it did not include any in situ validation of actual liquid water amount. Naderpour et al. (2021) supported Houtz et al. (2019) finding using a close-range (CR) single-angle L-band microwave radiometer measurements and the same L-band specific forward model (LS-MELMS) at the Swiss Camp. Houtz et al. (2021) extended the Houtz et al. (2019) approach to the entire GrIS. They tuned the wet layer thickness (10 cm – 100 cm) to provide variable estimates of liquid water which also were not validated against any reliable reference. Field observations and modeling results provide evidence of meltwater infiltration for more than 100 cm, especially in the percolation zone of the GrIS (e.g., Samimi et al. 2021; Vandecrux et al., 2020). Mousavi et al. (2021) developed an L-band specific snow/firn radiative transfer model to derive multidimensional brightness temperature look up tables for the frozen and melt season considering a four-layer ice sheet structure. The algorithm uses frozen season

brightness temperature to determine the baseline emissions (temperature, density, scattering coefficient) which are then used in melt season to estimate liquid water content and corresponding wet layer thickness. In this manuscript, we extend Mousavi et al., (2021) approach with improved and updated LUT to quantify and validate the LWA with two state-of-the-art surface energy balance models forced with in situ observations and reanalysis data products.

2. In equation (1) isn't this neglecting reflection?

Yes, Eq. 1 is a simple Rayleigh-Jeans approximation ignoring internal reflections and scattering, it is mentioned in lines 110-111 that "if firn were vertically homogeneous or isothermal", the TB could be found by this simple relation. But firn is neither isothermal nor homogeneous. So, TB are described by multi-layer radiative transfer (RT) equations (e.g., Eq. 2), which consider the depth-integrated product of physical temperature and emissivity, considering the emissive, absorptive, and scattering properties of the snow, firn, or ice layers. Eq. 1 was given as an introduction for general readers to clarify what radiometers measure and what the brightness temperature represents. But as the background discussion proceeds, actual snow/firn emission and radiative transfer with the emissive, absorptive, and scattering properties is clarified (lines 113-150).

3. Can you clarify in the RT model if multiple scattering or coherence are being considered or not? I see now in line 216 "incoherent approach", can you quickly qualitatively explain this assumption?

The RT model is incoherent and considering larger wavelength of L-band emissions (21 cm), we assume that this assumption does not pose a major issue. The model also does not calculate multiple scattering analytically. However, to account for the combined reflective effects by the complex stratigraphy due to numerous ice layers common in the percolation zone of the GrIS, as well as the effects of multiple scattering in the snow/firn layer, we consider a hypothetical highly reflective layer (Layer 2 underneath the dry/wet snow layer) by explicitly specifying its dielectric constant (with high real part that varies spatially).

4. Line 207, is there any justification to choosing the frozen and melt season reference dates? These would also vary with latitude and elevation.

There is no consensus in reference dates, however, Jan 1 – Mar 31 is generally considered fully frozen conditions regardless of elevation and latitudes. However, SMAP does not have data for Jan 1 – Mar 30, 2015, because the data production started on March 31, 2015; therefore, we extended the reference period to Apr 7th for all the years to make it consistent. October 24 – December 31 was determined based on visual observations of the time series during 2015-2023. We will add these clarifications in the revised manuscript.

5. I don't quite understand in line 215 how the top layer is considered infinite thickness and then discrete thickness layers are used beneath this? Maybe this just means that no attenuation is considered in these layers, only refraction, so it is independent of layer thickness?

The top layer is simply air above the snow and assumed to be semi-infinite (surface to radiometer antenna, top of the atmosphere at an altitude 685-km SMAP orbit).

6. In line 248, How is 5% determined to be the maximum volume fraction of water? What about large supraglacial lakes?

Experiments performed by Colbeck, (1974) suggest that because of capillary retention, the irreducible water saturation of dense snow/firn is 7% of its pore volume. Coléou and Lesaffre, (1998) showed that the irreducible water content can be up to 6.5 - 8.5 % of the pore volume depending on the density. Based on these studies, and considering snow/firn density in the percolation zone, we determined the maximum volume fraction to be 5%. Furthermore, we performed histogram analysis of model estimated liquid water content (Samimi et al., 2021 EBM model with in situ data) in the percolation zone to determine the maximum liquid water content. We found the range of volume fraction of liquid water is within 5%.

The focus of the manuscript is the percolation zone of Greenland ice sheet which does not include areas of large supraglacial lakes. However, we discussed the L-band radiometric responses of ablation zone where supraglacial lakes are located. Our detection algorithm can detect the presence of meltwater in the ablation zone, but it was not possible to quantify the amount of liquid water in these areas with the current algorithm. So, we masked the melt detection in these areas (Fig. 8). Supraglacial lakes and other intricate features in the ablation zone pose challenges for radiometric liquid water estimation. Due to soaked snow, the brightness temperature quickly saturates and decreases during melt season. We included discussion on this in lines 501-509 as a path forward.

7. Line 260. Can you explain "the inversion only considered increasing TBs for LWA quantification". How is "increasing" versus "decreasing" determined? From one daily average to the next? Does this assume that melt always increases brightness temperatures?

The increasing and decreasing are determined with respect to the frozen season reference threshold (T) as determined by Eq. 2 (i.e., mean frozen season TB plus/minus 10 times STD). Fig. 2 illustrates the areas where the TB increases (percolation zone) and decreases (ablation and higher accumulation zone) in the melt season due to melt. As shown, melt can both increase and decrease TB depending on the area (or amount of melt). Our detection algorithm considers both increasing and decreasing TBs during melt season to detect melt (any TB beyond threshold,

higher or lower, is considered as melt. See the sample detection in Fig. 2 indicated by blue square symbols). However, our LWC quantification algorithm only considers the increasing case in the percolation zone.

What about the reflective effect after water volume fraction becomes high and TB decreases again?

In the percolation zone, the assumption is that the LWC is constrained to the maximum volume fraction (5%), preventing the reflecting effects becoming dominant.

8. Line 363, "The AWS measurements that run the model" maybe "The AWS measurements for which the model was run" or "The AWS measurements that are the driving inputs of the model".

It will be revised as suggested.

9. Line 416 "2025 – 2023" I think you mean 2021-2023?

Yes, we meant 2021 – 2023. Sorry, it was a typo, and it will be corrected in the revised manuscript.

10. Line 498 mentions firn aquifers. What is the current status of identifying these? Will they be unidentifiable in the product because the frozen versus melt season TBs do not vary much? Are they apparent in the magnitude of the frozen season TBs? I believe this has been addressed to some degree in the literature, it might be helpful to add some references.

As we discussed in the lines 496-500, the detection algorithm follows a threshold-based technique that uses winter reference of the TB to detect melt events, while perennial firn aquifers store a large quantity of the LWA throughout the year. Miller et al. (2020, 2022b, a, 2023) represent the current status of identifying them.

The signal used to identify the aquifers during the frozen season in the Miller et al. studies is an order of magnitude smaller than what the melt signal is. Therefore, the firn aquifers do not affect the detection and quantification of the meltwater amounts.

We cited firn aquifer detection and modelling studies and will review the literature again to makes sure all relevant studies are included.

11. In addition to the technical limitations mentioned in the conclusion, I believe it could show great impact to also discuss longer-term plans and potential of this dataset. E.g. Is there any plan to provide integrated melt-water estimates across the ice sheet or use these retrievals in a bigger picture Surface Mass Balance study? Is there a path or plan towards generating a SMAP data product based on this algorithm?

We agree to and appreciate the reviewer's perspective that we may discuss more about our longer-term plans and potential of this dataset. The initial results included in the manuscript demonstrate SMAP's capability to measure both surface and subsurface meltwater across the percolation zone of the Greenland ice sheet, offering deeper insights into sub-surface conditions. Our long-term goal is to create a liquid water amount data product across GrIS, which can be used in GrIS surface mass balance study, including process level problems such as meltwater generation, retention (refreezing), and runoff, ice sheet and glacier surface evolution, atmospheric feedback etc. It can also be used to better constrain and validate regional climate models.

To provide a longer time data product, we are currently working to integrate SMOS observations (2010 - present) with SMAP. A detailed sensitivity analysis and uncertainty characterization of the LWA retrieval algorithm, including dielectric mixing models is underway. We are also investigating the added benefits of other complementary frequencies (6 GHz up to 36 GHz bands), aiming at CIMR (ESA's Copernicus Imaging Microwave Radiometer to be launched in 2029) which will include coincident L (1.4 GHz) - Ka (36 GHz) channels (Colliander et al., 2024; Kilic et al., 2018), for the first time, to provide a possible depth profile of the LWC. The algorithm can also be extended for LWA estimation in Antarctic Ice Sheet.

At the moment the SMAP mission has not decided to generate an operational meltwater product but that may change in the future.

A brief discussion on these future research directions will be added in the conclusion of the revised manuscript.

References:

Colbeck, S. C.: The capillary effects on water percolation in homogeneous snow, J. Glaciol., 13, 85–97, https://doi.org/10.3189/s002214300002339x, 1974.

Coléou, C. and Lesaffre, B.: Irreducible water saturation in snow: experimental results in a cold laboratory, Ann. Glaciol., 26, 64–68, https://doi.org/10.3189/1998aog26-1-64-68, 1998.

Colliander, A., Hossan, A., Harper, J., Vandecrux, B., Miller, J., Marshall, S., and Donlon, C.: Towards Ice Sheet and Ice Shelf Meltwater Profile Retrieval from Copernicus Imaging Microwave Radiometer ( CIMR ), 4–5, 2024.

Houtz, D., Naderpour, R., Schwank, M., and Steffen, K.: Snow wetness and density retrieved from L-band satellite radiometer observations over a site in the West Greenland ablation zone, Remote Sens. Environ., 235, 111361, https://doi.org/10.1016/j.rse.2019.111361, 2019.

Houtz, D., Mätzler, C., Naderpour, R., Schwank, M., and Steffen, K.: Quantifying Surface Melt and Liquid Water on the Greenland Ice Sheet using L-band Radiometry, Remote Sens. Environ., 256, https://doi.org/10.1016/j.rse.2021.112341, 2021.

Kilic, L., Prigent, C., Aires, F., Boutin, J., Heygster, G., Tonboe, R. T., Roquet, H., Jimenez, C., and Donlon, C.: Expected Performances of the Copernicus Imaging Microwave Radiometer (CIMR) for an All-Weather and High Spatial Resolution Estimation of Ocean and Sea Ice Parameters, J. Geophys. Res. Ocean., 123, 7564–7580, https://doi.org/10.1029/2018JC014408, 2018.

Miller, J. Z., Long, D. G., Jezek, K. C., Johnson, J. T., Brodzik, M. J., Shuman, C. A., Koenig, L. S., and Scambos, T. A.: Brief communication: Mapping Greenland's perennial firn aquifers using enhanced-resolution L-band brightness temperature image time series, Cryosphere, 14, 2809–2817, https://doi.org/10.5194/tc-14-2809-2020, 2020.

Miller, J. Z., Culberg, R., Long, D. G., Shuman, C. A., Schroeder, D. M., and Brodzik, M. J.: An empirical algorithm to map perennial firn aquifers and ice slabs within the Greenland Ice Sheet using satellite L-band microwave radiometry, Cryosphere, 16, 103–125, https://doi.org/10.5194/tc-16-103-2022, 2022a.

Miller, J. Z., Long, D. G., Shuman, C. A., Culberg, R., Hardman, M., and Brodzik, M. J.: Mapping Firn Saturation Over Greenland Using NASA's Soil Moisture Active Passive Satellite, IEEE J. Sel. Top. Appl. Earth Obs. Remote Sens., 15, 3714–3729, https://doi.org/10.1109/JSTARS.2022.3154968, 2022b.

Miller, J. Z., Long, D. G., Shuman, C. A., and Scambos, T. A.: Satellite Mapping of the Extent and Physical Characteristics of an Expansive Perennial Firn Aquifer in the Wilkins Ice Shelf, Antarctic Peninsula, 219–222, https://doi.org/10.1109/igarss52108.2023.10281647, 2023.

Mousavi, M., Colliander, A., Miller, J. Z., Entekhabi, D., Johnson, J. T., Shuman, C. A., Kimball, J. S., and Courville, Z. R.: Evaluation of Surface Melt on the Greenland Ice Sheet Using SMAP L-Band Microwave Radiometry, IEEE J. Sel. Top. Appl. Earth Obs. Remote Sens., 14, 11439–11449, https://doi.org/10.1109/JSTARS.2021.3124229, 2021.

Naderpour, R., Houtz, D., and Schwank, M.: Snow wetness retrieved from close-range L-band radiometry in the western Greenland ablation zone, J. Glaciol., 67, 27–38, https://doi.org/10.1017/jog.2020.79, 2021.

---

## Author Comment (AC2)

**Author Response to Reviewer #2** (Author responses are highlighted in orange)

General comments:

This work continues the author's previous research on the liquid water over Greenland. A radiative transfer model is implemented to generate a Lookup table to retrieve liquid water content during the melt season, and the retrieved results are compared with modeling results. The retrieved LWA shows a good agreement with the modeled results. This paper is of good quality, but I would like to ask a few questions before publication.

The authors would like to thank the anonymous reviewer for the detailed review and insightful comments.

Specific comments:

1. SMAP data sets. The authors chose the high-resolution SMAP data. What is the benefit of this data set in this research? In the discussion section, could the authors also elaborate more on the sentence" … overlapping observations to produce the 3.125 km gridded data but still has an effective spatial resolution of ~30 km"? How would this affect the results?

   The benefit of this data is that it improves the overall effective resolution. The effective resolution of the conventionally gridded TB images is approximately 46 km, and the effective resolution of the enhanced-resolution TB images is approximately 30 km resulting an improvement of about 30 % (Long et al., 2023). The 3.125 km is the posting grid where each grid cell is reconstructed using a set of $T_B$ observations that overlap within the 3.125 km grid box (i.e., they are not independent), enabling added spatial pattern fidelity compared to coarser grid postings, also discussed in Long et al. (2023) and Zeiger et al. (2024).

   The results are analyzed assuming the 30-km resolution; otherwise, it does not affect the meltwater amount retrieval results.

   We will make this description clearer in the data section of the revised manuscript.

2. Permittivity of the wet firn. The author mentioned equation (3), and it seems the authors are using this equation to calculate penetration depth and absorption coefficients in the radiative transfer equation. Could the author discuss the equation used for calculating the effective permittivity?

   The dielectric constant of the dry snow was calculated using Mätzler (2006), and wet snow following Ulaby and Long (2014). Ulaby and Long (2014) model of wet snow dielectric constant is  an empirical model, called the 'modified Debye-like model,' which

is an extension of Hallikainen et al. (1986). We did not include the equations in the manuscript as we used the exact formulation from Mätzler (2006) and Ulaby and Long (2014) without any modifications. We briefly mentioned about the dielectric models in lines 244-245, but we will expand it to clarify more in the revised version of the manuscript. Nevertheless, readers are referred to these references for more details.

3.  Penetration depth of the wet firn. What would be a typical number and range of the penetration depth?

    It depends on the average volume fraction of melt (liquid water content) and the dry snow/firn density. For a typical snow density (measured for dry snow) in the percolation zone, it can be more than 4 m for an average LWC of less than 1% with the Ulaby and Long (2014) model's wet snow dielectric constant, decreasing exponentially with the LWC. Thus, for an average LWC of 3% and higher, it is around 1 m and less (see the Figure R1 below). The average LWC in the percolation zone is typically not higher than 4%, except for extraordinary melt years (like 2012, not included in the study). The water percolation is also generally within upper 4 m.

[Figure]

Figure R1. Penetration depth of L-band (1.41 GHz) signals as a function of liquid water content for different snow densities.

4.  Layering model for the wet firn. Usually, the density profile in the percolation zone increases much faster than in the accumulation zone; the density can be close to the ice after a few meters, say 10m. I'm wondering how the authors deal with the varying density profile in the radiative transfer modeling, given that the penetration depth of 1% wet snow can be 10 meters based on the code provided in Microwave radar and radiometric remote sensing (by Ulaby and Long)?

    The density profile is highly variable in the percolation zone and while the ice can start at relatively shallow depths at some lower altitude locations, the refreezing of seasonal melt forming discrete ice layers and ice pipes that cause significant scattering of frozen season microwave emission down to tens of meters is more typical (e.g., Rennermalm et al., 2022). Fig. 1 illustrates the effects of this variability in the percolation zone with low brightness temperature, while the areas with solid ice (ablation zone) are characterized with high brightness temperatures.

    As discussed in Sec 2.3, to account for the combined reflective effects by the complex stratigraphy due to numerous ice layers common in the percolation zone of the GrIS, as well as the effects of volume scattering in the snow/firn layer, we consider layer 2 (underneath the dry/wet snow layer) as a highly reflective layer by explicitly specifying its dielectric constant (with high real part that varies spatially). We match simulated TB under frozen condition with the mean frozen season observed TB to adjust the value for the dielectric constant of this layer, as well as the dry snow/firn density and physical temperature of the top layer. These background conditions are then used to determine the liquid water amount by a second inversion with the observed TB in the melt season. We will revise Sec. 2.3 it to make clearer.

5.  Melt onset. The paper compares the melt onset with the modeling results. Could the auto weather stations provide any information (such as temperature) that can provide some ground truth evidence on the melt onset?

    The PROMICE/GC-Net automatic weather stations (AWS) provide 2m-air temperature. However, there have been several studies (e.g. Zhang et al., 2023; Leduc-Leballeur et al., 2020; Van Den Broeke et al., 2010) that showed that air temperature is not sufficient to validate the onset or presence of melt – there can be melt even when the air temperature is below the freezing point (0°C). Additionally, since the liquid water amount is presented on daily basis, there is an additional sampling issue of potential negative average air temperature on days when melt may occur during a short period, while the air temperature-based method records zero melt. Therefore, a sub-freezing point temperature is often recommended (Zhang et al., 2023; Leduc-Leballeur et al., 2020; Van Den Broeke

et al., 2010). However, there is no consensus about the threshold. Hence, we preferred to compare the melt onset detection to the surface energy balance (SEB) based method (the model) that considers the net energy balance at the surface coupled with the vertical heat transfer and water percolation in addition to the air temperature. The SEB based method is considered superior to the conventional air temperature based method (Ohmura, 2001).

6. Better correlation between SMAP and EBM. In Table 1, the comparison between SMAP and EBM seems to be in agreement. Why is this so? Is this because EBM is using AWS data for calibration? If so, can GEMB somehow be calibrated by the AWS?

Yes, we believe the overall better agreement between SMAP and EBM estimated LWA than SMAP and GEMB partly come from the surface forcing – AWS in situ measurements are better than ERA5 reanalysis. However, two model have different parametrizations of key physical processes such as snow densification, meltwater infiltration, thermal and hydraulic conductivity, permeability, grain growth, albedo etc. schemes. So, these processes need to be considered too to account for the differences which, as we discussed in Sec. 4, are out of the scope this article (some of these issues were partially covered by Vandecrux et al. (2020)). However, we are considering running both the model with the same input set (AWS data). Currently, GEMB setting is tied to ERA5 for large scale simulation. We are working to run it with point (AWS) measurements, and we anticipate we will be able to include it in the revised manuscript.

7. Possible overestimation of LWA by models. Figure 9 shows the comparison of measured and modeled temperature profiles. The modeled results seem to overestimate the temperatures and, thus, possibly the LWA. Any ideas on resolving this issue and validating the solution?

Yes, in some cases, the models seem to retain the subsurface meltwater with a persistent wetting front (0°C isotherm) for an extended period in late summer seasons. This overestimates the total LWA (or vice versa). As we discussed in lines 466-479: "Speculating extra melt production due to possible error in the AWS surface forcing, and other surface processes in the EBM, we examined modelled subsurface temperature profile by reducing surface melt with different factors (<1). We also performed similar analysis with irreducible water content, thermal conductivity. In either case (not shown), we could not match the subsurface profiles with measured profiles within reasonable agreements." Heat transfer and firn deification are common problems in firn models (Vandecrux et al., 2020). But there are other processes in the model (such as mentioned in 6) that may contribute to this. The AWS measurements used to run the model also add

some inherent uncertainties. Therefore, these problems are multifaceted, and additional works are required to understand the basis for these discrepancies.

**References:**

Van Den Broeke, M., Bus, C., Ettema, J., and Smeets, P.: Temperature thresholds for degree-day modelling of Greenland ice sheet melt rates, Geophys. Res. Lett., 37, 1–5, https://doi.org/10.1029/2010GL044123, 2010.

Christian Mätzler: Microwave dielectric properties of ice, Transportation (Amst)., 1, 21–30, https://doi.org/10.1002/ejoc.201200111, 2006.

Hallikainen, M. T., Ulaby, F. T., and Abdelrazik, M.: Dielectric Properties of Snow in the 3 To 37 Ghz Range., IEEE Trans. Antennas Propag., AP-34, 1329–1340, https://doi.org/10.1109/tap.1986.1143757, 1986.

Leduc-Leballeur, M., Picard, G., MacElloni, G., Mialon, A., and Kerr, Y. H.: Melt in Antarctica derived from Soil Moisture and Ocean Salinity (SMOS) observations at L band, Cryosphere, 14, 539–548, https://doi.org/10.5194/tc-14-539-2020, 2020.

Long, D. G., Brodzik, M. J., and Hardman, M. A.: Enhanced-Resolution SMAP Brightness Temperature Image Products, IEEE Trans. Geosci. Remote Sens., 57, 4151–4163, https://doi.org/10.1109/TGRS.2018.2889427, 2019.

Long, D. G., Brodzik, M. J., and Hardman, M.: Evaluating the effective resolution of enhanced resolution SMAP brightness temperature image products, Front. Remote Sens., 4, https://doi.org/10.3389/frsen.2023.1073765, 2023.

Ohmura, A.: Physical Basis for the Temperature-Based Melt-Index Method, J. Appl. Meteorol., 40, 753–761, 2001.

Rennermalm, Å. K., Hock, R., Covi, F., Xiao, J., Corti, G., Kingslake, J., Leidman, S. Z., Miège, C., Macferrin, M., Machguth, H., Osterberg, E., Kameda, T., & McConnell, J. R. (2021). Shallow firn cores 1989–2019 in southwest Greenland's percolation zone reveal decreasing density and ice layer thickness after 2012. In Journal of Glaciology (Vol. 68, Issue 269, pp. 431–442). Cambridge University Press (CUP). https://doi.org/10.1017/jog.2021.102

Ulaby, F. and Long, D.: Microwave Radar and Radiometric Remote Sensing, Microw. Radar Radiom. Remote Sens., https://doi.org/10.3998/0472119356, 2014.

Vandecrux, B., Mottram, R., L. Langen, P., S. Fausto, R., Olesen, M., Max Stevens, C., Verjans, V., Leeson, A., Ligtenberg, S., Kuipers Munneke, P., Marchenko, S., Van Pelt, W., R. Meyer, C., B. Simonsen, S., Heilig, A., Samimi, S., Marshall, S., MacHguth, H., MacFerrin, M., Niwano, M., Miller, O., I. Voss, C., and E. Box, J.: The firn meltwater Retention Model Intercomparison Project (RetMIP): Evaluation of nine firn models at four weather station sites on the Greenland ice sheet, Cryosphere, 14, 3785–3810, https://doi.org/10.5194/tc-14-3785-2020, 2020.

Zeiger, P., Picard, G., Richaume, P., Mialon, A., & Rodriguez-Fernandez, N. (2024). Resolution enhancement of SMOS brightness temperatures: Application to melt detection on the Antarctic

and Greenland ice sheets. In Remote Sensing of Environment (Vol. 315, p. 114469). Elsevier BV. https://doi.org/10.1016/j.rse.2024.114469

Zhang, Z., Zheng, L., Leng, W., Zhao, T., Li, T., and Liang, Q.: Toward a real validation of passive microwave snowmelt detection algorithms over the Antarctic Ice sheet, Int. J. Appl. Earth Obs. Geoinf., 125, 103600, https://doi.org/10.1016/j.jag.2023.103600, 2023.

---

## Author Response (AR1)

**Author Response to the Editor**

Dear Dr., Hossan and colleagues,

Many thanks for your detailed responses to the reviewers. The reviewers are both supportive of publication of this manuscript, which will be subject to minor revisions. Please proceed to make the changes indicated in response to the reviewers. Although not explicitly stated that this will be included in the text, please add the clarifying text in responses to reviewer 1 point 5.

Please could you also check the notation: is 'N' number of layers (line 119, 214) or something else (needs explanation) in line 206?

With best wishes,

Mel

Dear Editor,

Thank you very much for your review and decision. We made the following corrections:

- We added clarifying description of the model layers in Sec. 2.3 in responses to reviewer 1 point 5.
- In line 119, 'N' was meant as the number of plane-parallel layers used to describe the snowpack in general. So, it was not changed.
- In line 214 (and elsewhere), We replaced 'N' with specific number (4) of layers we used in the forward modelling to make it specific and clearer.
- In 214, We used m as a multiplier. Different studies used similar threshold with different values of m (so we keep the threshold equation in general form). However, we mentioned in the following line that we used m = 10 to derive the threshold.

In addition, we revised the manuscript according to reviewers' comments. Please, see a point-by-point reply to the comments below, and the marked-up manuscript version showing the changes made.

We appreciate your feedback. Please, let us know if you have any questions.

Sincerely,

Alamgir Hossan, on behalf of all the authors

**Author Response to Reviewer #1**(Author responses are highlighted in orange)

This manuscript introduces a highly relevant and impactful application of L-band radiometry in a relatively unexplored research field. The manuscript is generally of very high quality, well written, and with excellent figures. I recommend a few clarifications and comments below. I look forward to seeing the progress on this manuscript and the further development of this method (data product?) in the future.

The authors would like to thank the reviewer for the prompt and comprehensive review and thoughtful comments.

1. At the end of the introduction, you mention and cite a few previous attempts using passive microwave to quantify liquid water on the ice sheet. Can you briefly summarize the work to date and mention how the content of the manuscript adds to or differs from the current status of the subject?

In my opinion one of the primary novelties of the manuscript is the demonstration of liquid water retrieval with single-angle L-band satellite observations. You might also want to consider citing the below paper which also demonstrated single-incidence angle retrievals of snow liquid water but using a ground-based radiometer. *Naderpour, R., Houtz, D., & Schwank, M. (2021). Snow wetness retrieved from close-range L-band radiometry in the western Greenland ablation zone. Journal of Glaciology, 67(261), 27-38.*

We revised the relevant section in the introduction (lines 88-107 in the marked-up manuscript) to include the literature review given above, as follows:

Houtz et al. (2019) used the Soil Moisture and Ocean Salinity (SMOS) multi-angle L-band radiometric observations with a two-layer configuration of the L-band specific MEMLS (LS-MEMLS) model in an inversion-based retrieval framework for simultaneous estimation of snow liquid water content and density at the Swiss Camp site located in the ablation zone of the western Greenland ice sheet (GrIS). This initial study evaluated the results with in situ air temperature and another satellite-based empirical melt detection algorithm (the cross-polarized gradient ratio of 19 GHz and 37 GHz TBs); however, it did not include any in situ validation of actual liquid water amount. Naderpour et al. (2021) supported Houtz et al. (2019) finding using a close-range (CR) single-angle L-band microwave radiometer measurements and the same L-band specific forward model (LS-MELMS) at the Swiss Camp. Houtz et al. (2021) extended the Houtz et al. (2019) approach to the entire GrIS. They tuned the wet layer thickness (10 cm – 100 cm) to provide variable estimates of liquid water which also were not validated against any reliable reference. Field observations and modeling results provide evidence of meltwater infiltration for more than 100 cm, especially in the percolation zone of the GrIS (e.g., Samimi et al. 2021; Vandecrux et al., 2020). Mousavi et al. (2021) developed an L-band specific snow/firn radiative transfer model to derive multidimensional brightness temperature look up tables for the frozen and melt season considering a four-layer ice sheet structure. The algorithm uses frozen season

brightness temperature to determine the baseline emissions (temperature, density, scattering coefficient) which are then used in melt season to estimate liquid water content and corresponding wet layer thickness. In this manuscript, we extend Mousavi et al., (2021) approach with improved and updated LUT to quantify and validate the LWA with two state-of-the-art surface energy balance models forced with in situ observations and reanalysis data products.

2. In equation (1) isn't this neglecting reflection?

Yes, Eq. 1 is a simple Rayleigh-Jeans approximation ignoring internal reflections and scattering, it is mentioned in lines 110-111 that "if firn were vertically homogeneous or isothermal", the TB could be found by this simple relation. But firn is neither isothermal nor homogeneous. So, TB are described by multi-layer radiative transfer (RT) equations (e.g., Eq. 2), which consider the depth-integrated product of physical temperature and emissivity, considering the emissive, absorptive, and scattering properties of the snow, firn, or ice layers. Eq. 1 was given as an introduction for general readers to clarify what radiometers measure and what the brightness temperature represents. But as the background discussion proceeds, actual snow/firn emission and radiative transfer with the emissive, absorptive, and scattering properties is clarified (lines 113-150).

3. Can you clarify in the RT model if multiple scattering or coherence are being considered or not? I see now in line 216 "incoherent approach", can you quickly qualitatively explain this assumption?

The RT model is incoherent and considering larger wavelength of L-band emissions (21 cm), we assume that this assumption does not pose a major issue. The model also does not calculate multiple scattering analytically. However, to account for the combined reflective effects by the complex stratigraphy due to numerous ice layers common in the percolation zone of the GrIS, as well as the effects of multiple scattering in the snow/firn layer, we consider a hypothetical highly reflective layer (Layer 2 underneath the dry/wet snow layer) by explicitly specifying its dielectric constant (with high real part that varies spatially).

4. Line 207, is there any justification to choosing the frozen and melt season reference dates? These would also vary with latitude and elevation.

There is no consensus in reference dates, however, Jan 1 – Mar 31 is generally considered fully frozen conditions regardless of elevation and latitudes. However, SMAP does not have data for Jan 1 – Mar 30, 2015, because the data production started on March 31, 2015; therefore, we extended the reference period to Apr 7th for all the years to make it consistent. October 24 – December 31 was determined based on visual observations of the time series during 2015-2023. We added these clarifications in Sec. 2.3 of the revised manuscript (lines 234-237 in the marked-up manuscript).

5. I don't quite understand in line 215 how the top layer is considered infinite thickness and then discrete thickness layers are used beneath this? Maybe this just means that no attenuation is considered in these layers, only refraction, so it is independent of layer thickness?

The top layer is simply air above the snow and assumed to be semi-infinite (surface to radiometer antenna, top of the atmosphere at an altitude 685-km SMAP orbit).

We added a clearer description of the model layers in Sec. 2.3 (lines 244-250 in the marked-up revised manuscript).

6. In line 248, How is 5% determined to be the maximum volume fraction of water? What about large supraglacial lakes?

Experiments performed by Colbeck, (1974) suggest that because of capillary retention, the irreducible water saturation of dense snow/firn is 7% of its pore volume. Coléou and Lesaffre, (1998) showed that the irreducible water content can be up to 6.5 - 8.5 % of the pore volume depending on the density. Based on these studies, and considering snow/firn density in the percolation zone, we determined the maximum volume fraction to be 5%. Furthermore, we performed histogram analysis of model estimated liquid water content (Samimi et al., 2021 EBM model with in situ data) in the percolation zone to determine the maximum liquid water content. We found the range of volume fraction of liquid water is within 5%.  These justifications are added to the revised manuscript (lines 278-288 in the marked-up revised manuscript).

The focus of the manuscript is the percolation zone of Greenland ice sheet which does not include areas of large supraglacial lakes. However, we discussed the L-band radiometric responses of ablation zone where supraglacial lakes are located. Our detection algorithm can detect the presence of meltwater in the ablation zone, but it was not possible to quantify the amount of liquid water in these areas with the current algorithm. So, we masked the melt detection in these areas (Fig. 8). Supraglacial lakes and other intricate features in the ablation zone pose challenges for radiometric liquid water estimation. Due to soaked snow, the brightness temperature quickly saturates and decreases during melt season. We included discussion on this in lines 501-509 as a path forward.

7. Line 260. Can you explain "the inversion only considered increasing TBs for LWA quantification". How is "increasing" versus "decreasing" determined? From one daily average to the next? Does this assume that melt always increases brightness temperatures?

The increasing and decreasing are determined with respect to the frozen season reference threshold (T) as determined by Eq. 2 (i.e., mean frozen season TB plus/minus 10 times STD). Fig. 2 illustrates the areas where the TB increases (percolation zone) and decreases (ablation and higher accumulation zone) in the melt season due to melt. As shown, melt can both increase and decrease TB depending on the area (or amount of melt). Our detection algorithm considers both increasing and decreasing TBs during melt season to detect melt (any TB beyond threshold,

higher or lower, is considered as melt. See the sample detection in Fig. 2 indicated by blue square symbols). However, our LWC quantification algorithm only considers the increasing case in the percolation zone. Clarification about the reference is added to the manuscript (lines 302-303 in the marked-up revised manuscript).

What about the reflective effect after water volume fraction becomes high and TB decreases again?

In the percolation zone, the assumption is that the LWC is constrained to the maximum volume fraction (5%), preventing the reflecting effects becoming dominant.

8. Line 363, "The AWS measurements that run the model" maybe "The AWS measurements for which the model was run" or "The AWS measurements that are the driving inputs of the model".

The relevant sentence has been revised.

9. Line 416 "2025 – 2023" I think you mean 2021-2023?

Yes, we meant 2021 – 2023. Sorry, it was a typo, and it has been corrected in the revised manuscript.

10. Line 498 mentions firn aquifers. What is the current status of identifying these? Will they be unidentifiable in the product because the frozen versus melt season TBs do not vary much? Are they apparent in the magnitude of the frozen season TBs? I believe this has been addressed to some degree in the literature, it might be helpful to add some references.

As we discussed in the lines 496-500, the detection algorithm follows a threshold-based technique that uses winter reference of the TB to detect melt events, while perennial firn aquifers store a large quantity of the LWA throughout the year. Miller et al. (2020, 2022b, a, 2023) represent the current status of identifying them.

The signal used to identify the aquifers during the frozen season in the Miller et al. studies is an order of magnitude smaller than what the melt signal is. Therefore, the firn aquifers do not affect the detection and quantification of the meltwater amounts.

We added a brief explanation regarding detecting and quantifying liquid water in firn aquifer and relevant studies are included (lines 548-552 in the marked-up revised manuscript).

11. In addition to the technical limitations mentioned in the conclusion, I believe it could show great impact to also discuss longer-term plans and potential of this dataset. E.g. Is there any plan to provide integrated melt-water estimates across the ice sheet or use these retrievals in a bigger picture Surface Mass Balance study? Is there a path or plan towards generating a SMAP data product based on this algorithm?

We agree to and appreciate the reviewer's perspective that we may discuss more about our longer-term plans and potential of this dataset. The initial results included in the manuscript demonstrate SMAP's capability to measure both surface and subsurface meltwater across the percolation zone of the Greenland ice sheet, offering deeper insights into sub-surface conditions. Our long-term goal is to create a liquid water amount data product across GrIS, which can be used in GrIS surface mass balance study, including process level problems such as meltwater generation, retention (refreezing), and runoff, ice sheet and glacier surface evolution, atmospheric feedback etc. It can also be used to better constrain and validate regional climate models.

To provide a longer time data product, we are currently working to integrate SMOS observations (2010 - present) with SMAP. A detailed sensitivity analysis and uncertainty characterization of the LWA retrieval algorithm, including dielectric mixing models is underway. We are also investigating the added benefits of other complementary frequencies (6 GHz up to 36 GHz bands), aiming at CIMR (ESA's Copernicus Imaging Microwave Radiometer to be launched in 2029) which will include coincident L (1.4 GHz) - Ka (36 GHz) channels (Colliander et al., 2024; Kilic et al., 2018), for the first time, to provide a possible depth profile of the LWC. The algorithm can also be extended for LWA estimation in Antarctic Ice Sheet.

At the moment the SMAP mission has not decided to generate an operational meltwater product but that may change in the future.

A brief discussion on these future research directions has been added in the conclusion of the revised manuscript.

**Author Response to Reviewer #2** (Author responses are highlighted in orange)

General comments:

This work continues the author's previous research on the liquid water over Greenland. A radiative transfer model is implemented to generate a Lookup table to retrieve liquid water content during the melt season, and the retrieved results are compared with modeling results. The retrieved LWA shows a good agreement with the modeled results. This paper is of good quality, but I would like to ask a few questions before publication.

The authors would like to thank the anonymous reviewer for the detailed review and insightful comments.

Specific comments:

1. SMAP data sets. The authors chose the high-resolution SMAP data. What is the benefit of this data set in this research? In the discussion section, could the authors also elaborate

more on the sentence" … overlapping observations to produce the 3.125 km gridded data but still has an effective spatial resolution of ~30 km"? How would this affect the results?

The benefit of this data is that it improves the overall effective resolution. The effective resolution of the conventionally gridded TB images is approximately 46 km, and the effective resolution of the enhanced-resolution TB images is approximately 30 km resulting an improvement of about 30 % (Long et al., 2023). The 3.125 km is the posting grid where each grid cell is reconstructed using a set of $T_B$ observations that overlap within the 3.125 km grid box (i.e., they are not independent), enabling added spatial pattern fidelity compared to coarser grid postings, also discussed in Long et al. (2023) and Zeiger et al. (2024).

The results are analyzed assuming the 30-km resolution; otherwise, it does not affect the meltwater amount retrieval results.

We made this description clearer in the data section of the revised manuscript (lines 122-125 in the marked-up revised manuscript).

2.  Permittivity of the wet firn. The author mentioned equation (3), and it seems the authors are using this equation to calculate penetration depth and absorption coefficients in the radiative transfer equation. Could the author discuss the equation used for calculating the effective permittivity?

The dielectric constant of the dry snow was calculated using Mätzler (2006), and wet snow following Ulaby and Long (2014). Ulaby and Long (2014) model of wet snow dielectric constant is an empirical model, called the 'modified Debye-like model,' which is an extension of Hallikainen et al. (1986). We did not include the equations in the manuscript as we used the exact formulation from Mätzler (2006) and Ulaby and Long (2014) without any modifications. We briefly mentioned about the dielectric models in lines 244-245, but we expanded it (lines 284-287 in the marked-up version) to clarify more in the revised version of the manuscript. Nevertheless, readers are referred to these references for more details.

3.  Penetration depth of the wet firn. What would be a typical number and range of the penetration depth?

It depends on the average volume fraction of melt (liquid water content) and the dry snow/firn density. For a typical snow density (measured for dry snow) in the percolation zone, it can be more than 4 m for an average LWC of less than 1% with the Ulaby and Long (2014) model's wet snow dielectric constant, decreasing exponentially with the

LWC. Thus, for an average LWC of 3% and higher, it is around 1 m and less (see the Figure R1 below). The average LWC in the percolation zone is typically not higher than 4%, except for extraordinary melt years (like 2012, not included in the study). The water percolation is also generally within upper 4 m.

We added these typical numbers in the revised manuscript (lines 244-250 in the marked-up version of the manuscript).

[Figure]

Figure R1. Penetration depth of L-band (1.41 GHz) signals as a function of liquid water content for different snow densities.

4. Layering model for the wet firn. Usually, the density profile in the percolation zone increases much faster than in the accumulation zone; the density can be close to the ice after a few meters, say 10m. I'm wondering how the authors deal with the varying density profile in the radiative transfer modeling, given that the penetration depth of 1% wet snow can be 10 meters based on the code provided in Microwave radar and radiometric remote sensing (by Ulaby and Long)?

The density profile is highly variable in the percolation zone and while the ice can start at relatively shallow depths at some lower altitude locations, the refreezing of seasonal melt forming discrete ice layers and ice pipes that cause significant scattering of frozen season

microwave emission down to tens of meters is more typical (e.g., Rennermalm et al., 2022). Fig. 1 illustrates the effects of this variability in the percolation zone with low brightness temperature, while the areas with solid ice (ablation zone) are characterized with high brightness temperatures.

As discussed in Sec 2.3, to account for the combined reflective effects by the complex stratigraphy due to numerous ice layers common in the percolation zone of the GrIS, as well as the effects of volume scattering in the snow/firn layer, we consider layer 2 (underneath the dry/wet snow layer) as a highly reflective layer by explicitly specifying its dielectric constant (with high real part that varies spatially). We match simulated TB under frozen condition with the mean frozen season observed TB to adjust the value for the dielectric constant of this layer, as well as the dry snow/firn density and physical temperature of the top layer. These background conditions are then used to determine the liquid water amount by a second inversion with the observed TB in the melt season. We revised Sec. 2.3 of the revised manuscript to make it clearer.

5. Melt onset. The paper compares the melt onset with the modeling results. Could the auto weather stations provide any information (such as temperature) that can provide some ground truth evidence on the melt onset?

The PROMICE/GC-Net automatic weather stations (AWS) provide 2m-air temperature. However, there have been several studies (e.g. Zhang et al., 2023; Leduc-Leballeur et al., 2020; Van Den Broeke et al., 2010) that showed that air temperature is not sufficient to validate the onset or presence of melt – there can be melt even when the air temperature is below the freezing point (0°C). Additionally, since the liquid water amount is presented on daily basis, there is an additional sampling issue of potential negative average air temperature on days when melt may occur during a short period, while the air temperature-based method records zero melt. Therefore, a sub-freezing point temperature is often recommended (Zhang et al., 2023; Leduc-Leballeur et al., 2020; Van Den Broeke et al., 2010). However, there is no consensus about the threshold. Hence, we preferred to compare the melt onset detection to the surface energy balance (SEB) based method (the model) that considers the net energy balance at the surface coupled with the vertical heat transfer and water percolation in addition to the air temperature. The SEB based method is considered superior to the conventional air temperature based method (Ohmura, 2001).

6. Better correlation between SMAP and EBM. In Table 1, the comparison between SMAP and EBM seems to be in agreement. Why is this so? Is this because EBM is using AWS data for calibration? If so, can GEMB somehow be calibrated by the AWS?

Yes, we believe the overall better agreement between SMAP and EBM estimated LWA than SMAP and GEMB partly come from the surface forcing – AWS in situ measurements are better than ERA5 reanalysis. However, two model have different parametrizations of key physical processes such as snow densification, meltwater infiltration, thermal and hydraulic conductivity, permeability, grain growth, albedo etc. schemes. So, these processes need to be considered too to account for the differences which, as we discussed in Sec. 4, are out of the scope this article (some of these issues were partially covered by Vandecrux et al. (2020)). However, we are considering running both the model with the same input set (AWS data). Currently, GEMB setting is tied to ERA5 for large scale simulation.

We forced both the model with PROMICE AWS measurements and updated the results in the revised manuscript. **Please, see the updated results in Fig. 6 and Table I and II**. In Table I, we also confine the performance metrics to summer season only (Jun 1 – Oct 31) every year. In the 'Mean Difference', polarity of the numbers has been corrected (reversed).
Relevant texts (lines 27-33 in the abstract, 344, 353-359 in Sec 2.5.2, in the Result section (3.1.2)) have accordingly been revised to incorporate the changes in GEMB forcing.

7. Possible overestimation of LWA by models. Figure 9 shows the comparison of measured and modeled temperature profiles. The modeled results seem to overestimate the temperatures and, thus, possibly the LWA. Any ideas on resolving this issue and validating the solution?

Yes, in some cases, the models seem to retain the subsurface meltwater with a persistent wetting front (0°C isotherm) for an extended period in late summer seasons. This overestimates the total LWA (or vice versa). As we discussed in lines 466-479: "Speculating extra melt production due to possible error in the AWS surface forcing, and other surface processes in the EBM, we examined modelled subsurface temperature profile by reducing surface melt with different factors (<1). We also performed similar analysis with irreducible water content, thermal conductivity. In either case (not shown), we could not match the subsurface profiles with measured profiles within reasonable agreements." Heat transfer and firn deification are common problems in firn models (Vandecrux et al., 2020). But there are other processes in the model (such as mentioned in 6) that may contribute to this. The AWS measurements used to run the model also add some inherent uncertainties. Therefore, these problems are multifaceted, and additional works are required to understand the basis for these discrepancies.

---

## Author Response (AR2)

Dear Dr., Sandells,

Thank you for your painstaking review and guidelines. We appreciate it. We addressed the comments as follows.

- *Before this is published, please could you clarify how the LUT derivation has changed from Mousavi et al., 2021.*
    - The main differences are in the range and resolutions of the background parameters as described in Section 2.3 (Melt Retrieval Algorithm). Mousavi et al., 2021 (and 2022) generated look-up-tables (LUTs) with a larger range for background temperature (110-265K) for highly reflective layer (Layer 2 in Fig. 4), and the semi-infinite ice layer (Layer 3 in Fig. 4) than we used here. We used a similar procedure to derive the LUTs, but with a higher background temperature in the lower end (≥200K), as described in Sec. 2.5 (lines 276-294 in the marked-up manuscript).

- *Lines 278-294 (marked up version) describes your methodology, but not what Mousavi et al. did. Is the difference in stratigraphy / LUT resolution or something else? Please insert this into the manuscript.*
    - Please, see above, and Mousavi et al. (2021, 2022)'s work, to the best of the authors' ability, are inserted.

- *I have not undertaken a complete proof-read of the document, but noticed the following, so to save you a small amount of time later, please could you correct the following (line numbers in marked up version)*
    - Thank you for your review. We have made the following corrections.

- *Minor changes:*
  *- Line 249: 'high reflective layer' -> 'highly reflective layer':* Corrected
  *- Line 546: 'Aa' -> 'As':* Corrected
  *- Line 551. Is this a validation of algorithm that is difficult, rather than detection per se?*
    - Thanks, we simplified and rephrased it in the following way,
      "Miller et al., (2022a, b) developed empirical technique to map Greenland's perennial firn aquifers with SMAP L-band brightness temperature; however,

without complementary observations of firn aquifers via other means such as radar sounding while the detection itself is ambiguous, the quantification would be more challenging."

References:

1. Mousavi, M., Colliander, A., Miller, J. Z., Entekhabi, D., Johnson, J. T., Shuman, C. A., Kimball, J. S., and Courville, Z. R.: Evaluation of Surface Melt on the Greenland Ice Sheet Using SMAP L-Band Microwave Radiometry, IEEE J. Sel. Top. Appl. Earth Obs. Remote Sens., 14, 11439–11449, https://doi.org/10.1109/JSTARS.2021.3124229, 2021.
2. Mousavi, M., Colliander, A., Miller, J., and Kimball, J. S.: A Novel Approach to Map the Intensity of Surface Melting on the Antarctica Ice Sheet Using SMAP L-Band Microwave Radiometry, IEEE J. Sel. Top. Appl. Earth Obs. Remote Sens., 15, 1724–1743, https://doi.org/10.1109/JSTARS.2022.3147430, 2022.

---

## Author Response (AR3)

Dear Dr. Sandells,

Thank you for your question. It is our pleasure to address it as follows.

- *Please could you clarify what the improvements are for your work over the Mousavi work, especially if you have a narrower range of applicability? It is not at all clear how this is an extension and an improvement over the Mousavi look up table.*

The primary motivation of this paper is to present a validation attempt for liquid water retrieval based on Mousavi's approach. Previous studies (Mousavi et al., 2021; 2022, and also Houtz et al., 2019; 2021) did not present a validation approach for the liquid water retrievals against any reliable reference. To the best of our knowledge, our comparison with the two state-of-the-art surface energy balance models (GEMB and Samimi et al., 2021) forced with in situ observations (PROMICE and GC-Net AWS) represents the first attempt to validate the liquid water retrieval.

Regarding the look-up table generation, upon reviewing the Mousavi algorithm, we realized that some values used in the look-up building were unphysical, such as deep ice sheet temperatures below 200 K. Therefore, we updated these parts and documented the changes accordingly in the paper.

Thank you for your time.

Sincerely,

Alamgir, on behalf of all the authors. 5/28/2025

---

## Author Response (AR4)

Dear Dr. Sandells,

Thank you. The following changes have been made and inserted in response to the editor's suggestion.

*"Thanks for your written response to my request for clarification. Perhaps there has been a technical issue, but the newly uploaded files appear to be the same as the previous files. Please could you include in the paper an explanation of the difference of the LUT developed in this publication i.e. why a new LUT was generated rather than using the original Mousavi version and what is the range and resolution of both LUT?"*

1. The specific contributions of the study, over previous studies including Mousavi et al. (2021), have been clarified and inserted in the Introduction Sec. (Lines 95-105, marked-up manuscript version).
2. The difference of the LUTs developed in this publication for the original Mousavi (2021) version is specified in lines 266-276 (marked-up manuscript version).

Thank you for your time.

Sincerely,

Alamgir, on behalf of all the authors. 6/68/2025

---

## Author Response (AR5)

Dear Editor,

Thank you for your consideration. We have uploaded the required files by adding the full affiliations of the authors in the manuscript as recommended. Please, let us know if anything is missing.

Thanks again for your time.

Sincerely,

Alamgir, on behalf of all the authors. 7/16/2025